# Reparameterized LLM Training via Orthogonal Equivalence Transformation

Zeju Qiu[1]  Simon Buchholz[1]  Tim Z. Xiao[1]  Maximilian Dax[1]  Bernhard Schölkopf[1]  Weiyang Liu[1,2,*]

[1]Max Planck Institute for Intelligent Systems, Tübingen   [2]The Chinese University of Hong Kong

## Abstract

While Large language models (LLMs) are driving the rapid advancement of artificial intelligence, effectively and reliably training these large models remains one of the field's most significant challenges. To address this challenge, we propose POET, a novel reParameterized training algorithm that uses Orthogonal Equivalence Transformation to optimize neurons. Specifically, POET reparameterizes each neuron with two learnable orthogonal matrices and a fixed random weight matrix. Because of its provable preservation of spectral properties of weight matrices, POET can stably optimize the objective function with improved generalization. We further develop efficient approximations that make POET flexible and scalable for training large-scale neural networks. Extensive experiments validate the effectiveness and scalability of POET in training LLMs.

## 1   Introduction

Recent years have witnessed the increasing popularity of large language models (LLMs) in various applications, such as mathematical reasoning [13] and program synthesis [3] and decision-making [77]. Current LLMs are typically pre-trained using enormous computational resources on massive datasets containing trillions of tokens, with each training run that can take months to complete. Given such a huge training cost, how to effectively and reliably train them poses significant challenges.

The *de facto* way for training LLMs is to directly optimize weight matrices with the Adam optimizer [37, 55]. While conceptually simple, this direct optimization can be computationally intensive (due to the poor scaling with model size) and requires careful hyperparameter tuning to ensure stable convergence. More importantly, its generalization can remain suboptimal even if the training loss is perfectly minimized [36]. To stabilize training and enhance generalization, various weight regularization methods [4, 10, 12, 47, 49, 79] and weight normalization techniques [28, 38, 39, 50, 52, 54] have been proposed. Most of these methods boil down to improving spectral properties of weight matrices (*i.e.*, singular values) either explicitly or implicitly. Intuitively, the spectral norm of a weight matrix (*i.e.*, the largest singular value) provides an upper bound on how much a matrix can amplify the input vectors, which connects to the generalization properties. In general, smaller spectral norms (*i.e.*, better smoothness) are considered to be associated with stronger generalization, which inspires explicit spectrum control [33, 59, 67, 79]. Theoretical results [6] also suggest that weight matrices with bounded spectrum can provably guarantee generalization. Given the importance of the spectral properties of weight matrices, *what prevents us from controlling them during LLM training?*

- **Inefficacy of spectrum control**: Existing spectrum control methods constrain only the largest singular value, failing to effectively regularizing the full singular value spectrum. Moreover, there is also no guarantee for spectral norm regularization to effectively control the largest singular value.

- **Computational overhead**: Both spectral norm regularization [79] and spectral normalization [59] require computing the largest singular value of weight matrices. Even with power iteration, this still adds a significant overhead to the training process, especially when training large neural networks. Additionally, spectral regularization does not scale efficiently with increasing model size.

---

[*]Project lead & Corresponding author      Project page: `spherelab.ai/poet`

39th Conference on Neural Information Processing Systems (NeurIPS 2025).

To achieve effective weight spectrum control without the limitations above, we propose POET, a reParameterized training algorithm that uses Orthogonal Equivalence Transformation to indirectly learn weight matrices. Specifically, POET reparameterizes a weight matrix $W \in \mathbb{R}^{m \times n}$ with $RW_0P$ where $W_0 \in \mathbb{R}^{m \times n}$ is a randomly initialized weight matrix, $R \in \mathbb{R}^{m \times m}$ and $P \in \mathbb{R}^{n \times n}$ are two orthogonal matrices. Instead of optimizing weight matrices directly, POET keeps the ran-

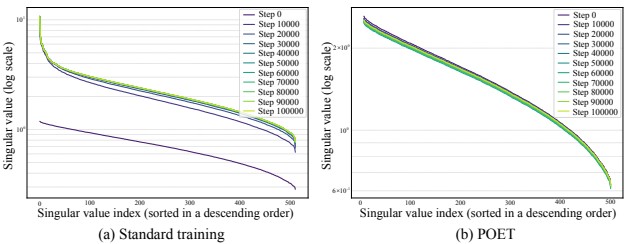

Figure 1: Training dynamics of singular values of the same weight matrix in a LLaMA model. Standard training on the left strictly follows the common practice for training LLMs (direct optimization with AdamW). POET on the right uses the proposed approximation for large-scale LLM training. The slight (almost negligible) singular value changes in POET are due to numerical and approximation error.

domly initialized weight matrix $W_0$ unchanged during training and learns two orthogonal matrices $R, P$ to transform $W_0$. This reparameterization preserves the singular values of weights while allowing flexible optimization of the singular vectors. POET effectively addresses the above limitations:

- **Strong spectrum control**: Because orthogonal transformations do not change the singular values of weight matrices, POET keeps the weight spectrum the same as the randomly initialized weight matrices (empirically validated by Figure 1 even with approximations). Through the initialization scheme, POET thus directly controls the singular value distribution of its weight matrices. As a result, and in contrast to standard LLM training, POET matrices avoid undesirable large singular values after training (Figure 1 and Appendix I). To further facilitate the POET algorithm, we introduce two new initialization schemes: normalized Gaussian initialization and uniform spectrum initialization, which can ensure the resulting weight matrices have bounded singular values.

- **Efficient approximation**: While a naive implementation of POET can be computationally expensive, its inherent flexibility opens up opportunities for efficient and scalable training. To address the key challenge of optimizing large orthogonal matrices, we introduce two levels of approximations:

  - *Stochastic primitive optimization*: The first-level approximation aims to reduce the number of learnable parameters when optimizing a large orthogonal matrix. To this end, we propose the stochastic primitive optimization (SPO) algorithm. Given a large orthogonal matrix $R \in \mathbb{R}^{m \times m}$, SPO factorizes it into a product of primitive orthogonal matrices, each involving significantly fewer trainable parameters. These primitives are constructed by parameterizing randomly sampled submatrices of the full matrix. This factorization is implemented as a memory-efficient iterative algorithm that sequentially updates one primitive orthogonal matrix at a time. To improve the expressiveness of the sequential factorization, we adopt a merge-then-reinitialize trick, where we merge each learned primitive orthogonal matrix into the weight matrix, and then reinitialize the primitive orthogonal matrix to be identity after every fixed number of iterations.

  - *Approximate orthogonality via Cayley-Neumann parameterization*: The second-level approximation addresses how to maintain orthogonality without introducing significant computational overhead. To achieve this, we develop the Cayley-Neumann parameterization (CNP) which approximates the Cayley orthogonal parameterization [48, 65] with Neumann series. Our merge-then-reinitialize trick can effectively prevent the accumulation of approximation errors.

POET can be viewed as a natural generalization of orthogonal training [48, 51, 65], wherein the model training is done by learning a layer-shared orthogonal transformation for neurons. Orthogonal training preserves the hyperspherical energy [47, 49] within each layer–a quantity that characterizes pairwise neuron relationships on the unit hypersphere. While preserving hyperspherical energy proves effective for many finetuning tasks [51], it limits the flexibility of pretraining. Motivated by this, POET generalizes energy preservation to spectrum preservation and subsumes orthogonal training as its special case. The better flexibility of POET comes from its inductive structures for preserving weight spectrum, rather than more learnable parameters. We empirically validate that POET achieves better pretraining performance than orthogonal training given the same budget of parameters.

To better understand how POET functions, we employ *vector probing* to analyze the learning dynamics of the orthogonal matrices. Vector probing evaluates an orthogonal matrix $R$ using a fixed, randomly generated unit vector $v$ by computing $v^\top R v$ which corresponds to the cosine similarity between $Rv$ and $v$. By inspecting the cosine similarities of seven orthogonal matrices throughout training, we

observe that the learning process can be divided into three distinct phases (Figure 2): (1) *conical shell searching*: The cosine starts at 1 (*i.e.*, $\boldsymbol{R}$ is the identity) and gradually converges to a stable range of $[0.6, 0.65]$, which we observe consistently across all learnable orthogonal matri-

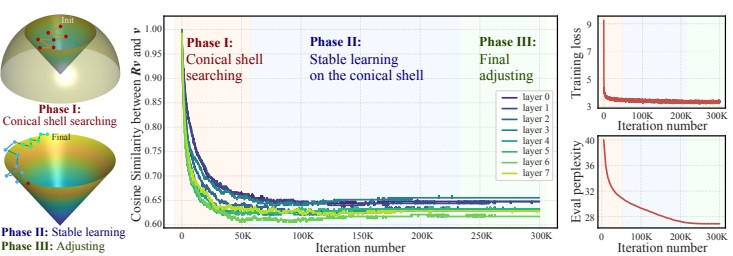

Figure 2: POET's three learning phases. Left: illustration; Middle: angle; Right: loss and validation.

ces. This suggests that $\boldsymbol{R}$ transforms $\boldsymbol{v}$ into a thin conical shell around its original direction. (2) *stable learning on the conical shell*: The cosine remains within this range while the model begins to learn stably. Despite the cosine plateauing, validation perplexity continues to improve almost linearly. (3) *final adjusting*: Learning slows and eventually halts as the learning rate approaches zero. We provide an in-depth discussion and empirical results in Appendix A,G. Our major contributions are:

- We introduce POET, a novel training framework that provably preserves spectral properties of weight matrices through orthogonal equivalence transformation.

- To enhance POET's scalability, we develop two simple yet effective approximations: stochastic principal submatrix optimization for large orthogonal matrices and the Cayley-Neumann parameterization for efficient representation of orthogonal matrices.

- We empirically validate POET's training stability and generalization across multiple model scales.

## 2 From Energy-preserving Training to Spectrum-preserving Training

Orthogonal training [48, 51, 65] is a framework to train neural networks by learning a layer-shared orthogonal transformation for neurons in each layer. For a weight matrix $\boldsymbol{W} = \{\boldsymbol{w}_1, \cdots, \boldsymbol{w}_n\} \in \mathbb{R}^{m \times n}$ where $\boldsymbol{w}_i \in \mathbb{R}^m$ is the $i$-th neuron, the layer's forward pass is $\boldsymbol{y} = \boldsymbol{W}^\top \boldsymbol{x}$ with input $\boldsymbol{x} \in \mathbb{R}^m$ and output $\boldsymbol{y} \in \mathbb{R}^n$. Unlike standard training, which directly optimizes $\boldsymbol{W}$, orthogonal training keeps $\boldsymbol{W}$ fixed at its random initialization $\boldsymbol{W}_0 = \boldsymbol{w}_1^0, \ldots, \boldsymbol{w}_n^0$ and instead learns an orthogonal matrix $\boldsymbol{R} \in \mathbb{R}^{m \times m}$ to jointly transform all neurons in the layer. The forward pass becomes $\boldsymbol{y} = (\boldsymbol{R}\boldsymbol{W}_0)^\top \boldsymbol{x}$. The effective weight matrix is $\boldsymbol{W}_R = \{\boldsymbol{w}_1^R, \cdots, \boldsymbol{w}_n^R\}$ where $\boldsymbol{w}_i^R = \boldsymbol{R}\boldsymbol{w}_i$. A key property is its *preservation of hyperspherical energy*. With $\hat{\boldsymbol{w}}_i = \boldsymbol{w}_i / \|\boldsymbol{w}_i\|$, orthogonal training ensures

$$\mathrm{HE}(\boldsymbol{W}_0) := \sum_{i \neq j} \left\| \hat{\boldsymbol{w}}_i^0 - \hat{\boldsymbol{w}}_j^0 \right\|^{-1} = \sum_{i \neq j} \left\| \boldsymbol{R}\hat{\boldsymbol{w}}_i - \boldsymbol{R}\hat{\boldsymbol{w}}_j \right\|^{-1} =: \mathrm{HE}(\boldsymbol{W}^R), \tag{1}$$

where hyperspherical energy $\mathrm{HE}(\cdot)$ characterizes the uniformity of neurons on a unit hypersphere. Prior work [47–49, 76] has shown that energy-preserving training can effectively improve generalization. Orthogonal finetuning (OFT) [51, 65] also demonstrates that finetuning foundation models while preserving hyperspherical energy achieves a favorable trade-off between efficient adaptation to downstream tasks and retention of pretraining knowledge. While the hyperspherical energy preservation is effective for finetuning, it can be too restrictive for pretraining. To allow greater flexibility in the pretraining phase, we relax the constraint from preserving hyperspherical energy to preserving the singular-value spectrum instead. By inherently maintaining the spectrum, energy-preserving training is a special case of spectrum-preserving training. As a generalization, spectrum-preserving training learns a transformation $\mathcal{T} : \mathbb{R}^{m \times n} \to \mathbb{R}^{m \times n}$ that preserves the spectrum:

$$\left\{ \sigma_1(\mathcal{T}(\boldsymbol{W}_0)), \sigma_2(\mathcal{T}(\boldsymbol{W}_0)), \cdots, \sigma_{\min(m,n)}(\mathcal{T}(\boldsymbol{W}_0)) \right\} = \left\{ \sigma_1(\boldsymbol{W}_0), \sigma_2(\boldsymbol{W}_0), \cdots, \sigma_{\min(m,n)}(\boldsymbol{W}_0) \right\}, \tag{2}$$

where $\sigma_i(\boldsymbol{W}_0)$ denotes the $i$-th singular value of $\boldsymbol{W}_0$ (sorted by descending order with $\sigma_1$ being the largest singular value). How we instantiate the transformation $\mathcal{T}$ results in different algorithms. Generally, $\mathcal{T}$ is a spectrum-preserving map, and can be either linear [42] or nonlinear [5]. If we only consider $\mathcal{T}$ to be a linear map, then Theorem 1 can fully characterize the form of $\mathcal{T}$:

**Theorem 1** (Simplified results from [42])**.** *For a linear map $\mathcal{T} : \mathbb{R}^{m \times n} \to \mathbb{R}^{m \times n}$ ($m \neq n$), if $\sigma_1(\mathcal{T}(\boldsymbol{W})) = \sigma_1(\boldsymbol{W})$ always holds for all $\boldsymbol{W} \in \mathbb{R}^{m \times n}$, then the linear map $\mathcal{T}$ must be of the following form: $\mathcal{T}(\boldsymbol{W}) = \boldsymbol{R}\boldsymbol{W}\boldsymbol{P}$, for all $\boldsymbol{W} \in \mathbb{R}^{m \times n}$ where $\boldsymbol{R} \in \mathbb{R}^{m \times m}$ and $\boldsymbol{P} \in \mathbb{R}^{n \times n}$ are some fixed elements in orthogonal groups $O(m)$ and $O(n)$, respectively.*

All parameterizations for the linear map $\mathcal{T}$ can be expressed as $\mathcal{T}(\boldsymbol{W}) = \boldsymbol{R}\boldsymbol{W}\boldsymbol{P}$, where $\boldsymbol{R}$ and $\boldsymbol{P}$ are orthogonal matrices. For instance, OFT is an energy-preserving method (a special case of spectrum-preserving training), where the map simplifies to $\mathcal{T}(\boldsymbol{W}) = \boldsymbol{R}\boldsymbol{W}\boldsymbol{I}$, with $\boldsymbol{I}$ as the identity.

**POET preserves hyperspherical energy under isotropic Gaussian initialization**. [48] shows that weight matrices that are initialized by zero-mean isotropic Gaussian distribution (*e.g.*, [17]) are guaranteed to have small hyperspherical energy. Because zero-mean isotropic Gaussian is invariant under orthogonal transformation, $RWP$ also has small energy (see Section 4 and Appendix B).

## 3 Reparameterized Training via Orthogonal Equivalence Transformation

This section introduces the POET framework, which reparameterizes each neuron as the product of a fixed random weight matrix and two learnable orthogonal matrices applied on both sides. POET serves as a specific implementation of spectrum-preserving training. Inspired by Theorem 1, it parameterizes the spectrum-preserving transformation $\mathcal{T}$ using a left orthogonal matrix that transforms the column space of the weight matrix and a right orthogonal matrix that transforms its row space.

### 3.1 General Framework

Following the general form of spectrum-preserving linear maps discussed in the last section, POET reparameterizes the neuron as $RW_0P$, where $W_0 \in \mathbb{R}^{m \times n}$ is a randomly initialized weight matrix that remains fixed during training, and $R \in \mathbb{R}^{m \times m}, P \in \mathbb{R}^{n \times n}$ are trainable orthogonal matrices. This reparameterization effectively applies an orthogonal equivalence transformation (OET) to random weight matrices. Specifically, OET is a double-sided transformation, defined as $\mathrm{OET}(W; R, P) = RWP$, where the input matrix $W$ is multiplied on the left and on the right by orthogonal matrices $R$ and $P$, respectively. The forward pass of POET can be thus written as

$$y = W_{RP}^\top x = (RW_0P)^\top x, \quad \text{s.t.} \left\{ R^\top R = RR^\top = I, \ P^\top P = PP^\top = I \right\}, \tag{3}$$

where $R$ and $P$ can be merged into a single weight matrix $W_{RP} = RW_0P$ after training. Therefore, the inference speed of POET-trained neural networks is the same as conventionally trained ones.

**Spectrum control**. POET can be interpreted as learning weight matrices by simultaneously transforming their left singular vectors and right singular vectors while keeping the singular values unchanged. Given the singular value decomposition (SVD) $W_0 = U\Sigma_0V^\top$, the reparameterized neuron weight matrix becomes $W_{RP} = RU\Sigma_0V^\top P$ where both $RU$ and $V^\top P$ are orthogonal matrices. This effectively constitutes an SVD of $W_{RP}$. It is also straightforward to verify that the spectral properties of $W_{RP}$ remain identical to those of the initial matrix $W_0$.

**Neuron initialization**. Since POET preserves the spectral properties of the initial weight matrix $W_0$, the choice of initialization plays a critical role. We consider two common schemes: (1) *standard initialization*, which samples from a zero-mean Gaussian with fixed variance (the default choice for LLaMA models); and (2) *Xavier initialization* [17], which uses a zero-mean Gaussian with variance scaled by the layer dimensions. To facilitate POET, we propose two new initialization schemes. The first method, *uniform-spectrum initialization*, applies SVD to a standard initialization and sets all singular values to 1, balancing spectral properties throughout training. The second, *normalized Gaussian initialization*, normalizes neurons drawn from a zero-mean Gaussian with fixed variance. This

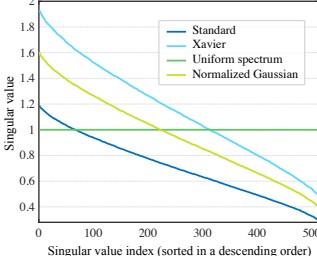

Figure 3: Singular values of a weight matrix of size $512 \times 1376$, randomly generated by different initialization schemes.

is directly inspired by prior work showing that normalized neurons improve convergence [48, 50, 52]. To ensure that the POET-reparameterized network is statistically equivalent to a standard network at initialization, we always initialize both orthogonal matrices as identity matrices.

### 3.2 Efficient Approximation to Orthogonality

POET is conceptually simple, requiring only the optimization of two orthogonal matrices. However, these matrices are typically large, and naively optimizing them leads to significant computational challenges. We start by introducing the following efficient approximations.

#### 3.2.1 Stochastic Primitive Optimization

The core idea of SPO is inspired by how QR factorization is performed using Givens rotations and Householder transformations. Both methods construct a large orthogonal matrix $R$ by sequentially applying primitive orthogonal transformations (*e.g.*, Givens rotations or Householder reflections), *i.e.*, $R = \prod_{i=1}^{c} G_i$, where $G_i$ denotes the $i$-th primitive orthogonal matrix. While each $G_i$ is of the same size as $R$, it is parameterized by significantly fewer degrees of freedom. See Figure 4

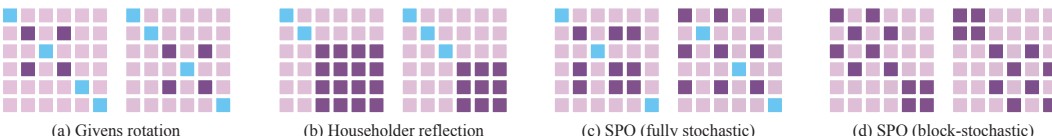

| (a) Givens rotation | (b) Householder reflection | (c) SPO (fully stochastic) | (d) SPO (block-stochastic) |

Figure 4: Examples of the primitive orthogonal transformation matrix $\boldsymbol{G}_i$ in different orthogonalizations (two examples for each method). Note that, blue blocks represent 1, light purple blocks denote 0 and deep purple blocks are the actual orthogonal parameterization to be learned.

for an illustration. Both Givens rotation and Householder reflection use relatively low-capacity parameterizations–for example, each Givens rotation $\boldsymbol{G}_i$ involves only a single effective parameter–which limits their efficiency in representing the full orthogonal matrix. SPO follows a similar idea of factorizing the original orthogonal matrix into multiple primitive orthogonal matrices. However, unlike Givens and Householder methods, SPO treats the number of effective parameters in each primitive matrix as a tunable hyperparameter and adopts a stochastic sparsity pattern.

**Fully stochastic SPO**. The basic idea of fully stochastic SPO is to randomly sample a small submatrix and enforce its orthogonality, allowing it to be easily extended to a full orthogonal matrix by embedding it within an identity matrix–a process similar to Givens or Householder transformations. To represent a large orthogonal matrix $\boldsymbol{R} \in \mathbb{R}^{m \times m}$, we start by defining $c$ index sets $\boldsymbol{S}^j = \{s_1^j, \cdots, s_b^j\} \subseteq \{1, \cdots, m\}$ ($j \in [1, c]$), where each set has cardinality $|\boldsymbol{S}^j| = b$, a hyperparameter controlling the number of effective parameters of a primitive orthogonal matrix. $\boldsymbol{S}^j, \forall j$ are randomly sampled from the full indices $\{1, \cdots, m\}$. Let $\tilde{\boldsymbol{G}}_j \in \mathbb{R}^{b \times b}$ be a small orthogonal matrix, and $\boldsymbol{D}(\boldsymbol{S}^j) = \{\boldsymbol{e}(s_1^j), \cdots, \boldsymbol{e}(s_b^j)\} \in \mathbb{R}^{m \times b}$ be a selection matrix, where $\boldsymbol{e}(k)$ is the standard basis vector with a 1 in the $k$-th position and 0 elsewhere. The factorization is given by

$$\boldsymbol{R} = \prod_{i=1}^{c} \big( \underbrace{\boldsymbol{I}_m + \boldsymbol{D}(\boldsymbol{S}^i) \cdot (\tilde{\boldsymbol{G}}_i - \boldsymbol{I}_b) \cdot \boldsymbol{D}(\boldsymbol{S}^i)^\top}_{\boldsymbol{G}_i:\ \text{The } i\text{-th primitive orthogonal matrix}} \big), \quad \text{s.t. } \tilde{\boldsymbol{G}}_i^\top \tilde{\boldsymbol{G}}_i = \tilde{\boldsymbol{G}}_i \tilde{\boldsymbol{G}}_i^\top = \boldsymbol{I}_b,\ \forall i, \tag{4}$$

where $\boldsymbol{D}(\boldsymbol{S}^i) \cdot (\boldsymbol{A}) \cdot \boldsymbol{D}(\boldsymbol{S}^i)^\top$ is a projector that replaces the $b \times b$ sub-block with $\boldsymbol{A}$. $\boldsymbol{I}_m$ and $\boldsymbol{I}_b$ are identity matrices of size $m \times m$ and $b \times b$, respectively. To efficiently parameterize small orthogonal matrices $\tilde{\boldsymbol{G}}_i$, we can use the CNP introduced in the next section.

**Block-stochastic SPO**. While fully stochastic SPO is simple, it may fail to transform all neuron dimensions because the identity matrix leaves part of the space unchanged. See the blue blocks in Figure 4(c) as an example. To address this, we propose block-stochastic SPO, which first constructs a block-diagonal orthogonal matrix with small blocks for parameter efficiency, and then applies a random permutation to enhance expressiveness by randomizing the sparsity pattern. Block-stochastic SPO transforms all neuron dimensions simultaneously, as shown in Figure 4(d). Formally we have

$$\boldsymbol{R} = \prod_{i=1}^{c} \big( \underbrace{\boldsymbol{\Psi}_i^\top \cdot \text{Diag}(\tilde{\boldsymbol{G}}_i^1, \tilde{\boldsymbol{G}}_i^2, \cdots, \tilde{\boldsymbol{G}}_i^{\lceil \frac{m}{b} \rceil}) \cdot \boldsymbol{\Psi}_i}_{\boldsymbol{G}_i:\ \text{The } i\text{-th primitive orthogonal matrix}} \big), \quad \text{s.t. } (\tilde{\boldsymbol{G}}_i^j)^\top \tilde{\boldsymbol{G}}_i^j = \tilde{\boldsymbol{G}}_i^j (\tilde{\boldsymbol{G}}_i^j)^\top = \boldsymbol{I}_b,\ \forall i, j, \tag{5}$$

where $\tilde{\boldsymbol{G}}_i^j \in \mathbb{R}^{b \times b}$ is the $j$-th block of the block diagonal matrix, and $\boldsymbol{\Psi}_i, \forall i$ are all random permutation matrices. As long as each diagonal block $\tilde{\boldsymbol{G}}_i^j$ is an orthogonal matrix, both $\boldsymbol{G}_i$ and $\boldsymbol{R}$ are also orthogonal matrices. We also use CNP to efficiently parameterize each orthogonal block $\tilde{\boldsymbol{G}}_i^j$.

**The merge-then-reinitialize trick**. The factorizations in Equation (4) and (5) offer a simple approach to optimizing large orthogonal matrices by sequentially updating primitive orthogonal matrices. However, storing all previous primitives incurs high GPU memory overhead. To mitigate this, we propose the merge-then-reinitialize trick, where the learned primitive orthogonal matrix can be merged into the weight matrix after every certain number of iterations, and then reinitialized to the identity matrix. After reinitialization, stochastic sampling is repeated to select a new index set (in fully stochastic SPO) or generate a new permutation (in block-stochastic SPO). This trick allows only one primitive matrix to be stored at a time, substantially reducing GPU memory usage.

### 3.2.2 Cayley-Neumann Parameterization

The classic Cayley parameterization generates an orthogonal matrix $\boldsymbol{R}$ in the form of $\boldsymbol{R} = (\boldsymbol{I} + \boldsymbol{Q})(\boldsymbol{I} - \boldsymbol{Q})^{-1}$ where $\boldsymbol{Q}$ is a skew-symmetric matrix satisfying $\boldsymbol{Q} = -\boldsymbol{Q}^\top$. A minor caveat of this parameterization is that it only produces orthogonal matrices with determinant 1 (*i.e.*, elements of the special orthogonal group), but empirical results in [48, 51, 65] indicate that this constraint does not hurt performance. However, the matrix inverse in the original Cayley parameterization introduces

numerical instability and computational overhead, limiting its scalability to large orthogonal matrices. To address this, we approximate the matrix inverse using a truncated Neumann series:

$$\boldsymbol{R} = (\boldsymbol{I} + \boldsymbol{Q})(\boldsymbol{I} - \boldsymbol{Q})^{-1} = (\boldsymbol{I} + \boldsymbol{Q}) \cdot \Big(\sum_{i=0}^{\infty} \boldsymbol{Q}^i\Big) \approx (\boldsymbol{I} + \boldsymbol{Q}) \cdot \Big(\sum_{i=0}^{k} \boldsymbol{Q}^i\Big) = (\boldsymbol{I} + \boldsymbol{Q}) \cdot \Big(\boldsymbol{I} + \sum_{i=1}^{k} \boldsymbol{Q}^i\Big), \quad (6)$$

where a larger number of approximation terms $k$ leads to a smaller approximation error. By avoiding matrix inversion, the training stability of POET is improved; however, this comes with a price–the approximation is valid only when the Neumann series converges in the operator norm. To initialize orthogonal matrices as identity, we set $\boldsymbol{Q}$ to a zero matrix in CNP, satisfying the convergence condition initially. As the training progresses, however, updates to $\boldsymbol{Q}$ may cause its operator norm to exceed $1$, violating this condition. Fortunately, our merge-then-reinitialize trick mitigates this issue by periodically resetting $\boldsymbol{Q}$ to a zero matrix, ensuring its operator norm remains small.

### 3.2.3 Overall Training Algorithm

**Step 1: Initialization**. We initialize the weight matrices using normalized Gaussian: $\boldsymbol{W} \leftarrow \boldsymbol{W}_0$.

**Step 2: Orthogonal matrix initialization**. For fully stochastic SPO, we randomly sample an index set $\boldsymbol{S}$, and parameterize $\tilde{\boldsymbol{G}}_R \in \mathbb{R}^{b \times b}$ and $\tilde{\boldsymbol{G}}_P \in \mathbb{R}^{b \times b}$ using CNP (Equation (6)). Both matrices are initialized as identity, so $\boldsymbol{R}$ and $\boldsymbol{P}$ also start as identity matrices. For block-stochastic SPO, we sample a random permutation matrix $\boldsymbol{\Psi}_R, \boldsymbol{\Psi}_P$, and parameterize $\{\tilde{\boldsymbol{G}}_R^1, \cdots, \tilde{\boldsymbol{G}}_R^{\lceil \frac{m}{b} \rceil}\}$ and $\{\tilde{\boldsymbol{G}}_P^1, \cdots, \tilde{\boldsymbol{G}}_P^{\lceil \frac{n}{b} \rceil}\}$ using CNP. Then we initialize them as the identity, so $\boldsymbol{R}$ and $\boldsymbol{P}$ again starts as identity matrices.

**Step 3: Efficient orthogonal parameterization**. For fully stochastic SPO, we have $\boldsymbol{R} = \boldsymbol{I}_m + \boldsymbol{D}(\boldsymbol{S})(\tilde{\boldsymbol{G}}_R - \boldsymbol{I}_b)\boldsymbol{D}(\boldsymbol{S})^{\top}$ and $\boldsymbol{P} = \boldsymbol{I}_n + \boldsymbol{D}(\boldsymbol{S})(\tilde{\boldsymbol{G}}_P - \boldsymbol{I}_b)\boldsymbol{D}(\boldsymbol{S})^{\top}$. For block-stochastic SPO, we have $\boldsymbol{R} = \boldsymbol{\Psi}_R^{\top}\text{Diag}(\tilde{\boldsymbol{G}}_R^1, \cdots, \tilde{\boldsymbol{G}}_R^{\lceil \frac{m}{b} \rceil})\boldsymbol{\Psi}_R$ and $\boldsymbol{P} = \boldsymbol{\Psi}_P^{\top}\text{Diag}(\tilde{\boldsymbol{G}}_P^1, \cdots, \tilde{\boldsymbol{G}}_P^{\lceil \frac{n}{b} \rceil})\boldsymbol{\Psi}_P$.

**Step 4: Inner training loop for updating orthogonal matrices**. The equivalent weight matrix in the forward pass is $\boldsymbol{R}\boldsymbol{W}\boldsymbol{P}$. Gradients are backpropagated through $\boldsymbol{R}$ and $\boldsymbol{P}$ to update $\tilde{\boldsymbol{G}}_R, \tilde{\boldsymbol{G}}_P$ (fully stochastic) or $\tilde{\boldsymbol{G}}_R^i, \tilde{\boldsymbol{G}}_P^i, \forall i$ (block-stochastic). This inner loop runs for a fixed number of iterations.

**Step 5: Merge-then-reinitialize**. The learned orthogonal matrices $\boldsymbol{R}$ and $\boldsymbol{P}$ are merged into the weight matrix by $\boldsymbol{W} \leftarrow \boldsymbol{R}\boldsymbol{W}\boldsymbol{P}$. If not terminated, return to **Step 2** for reinitialization.

## 4 Discussions and Intriguing Insights

**Parameter and memory complexity**. By introducing a hyperparameter $b$ as the sampling budget, fully stochastic SPO decouples parameter complexity from the size of the weight matrices. With a small $b$, POET becomes highly parameter-efficient, though at the cost of slower convergence. This offers users a flexible trade-off between efficiency and speed. In contrast, block-stochastic SPO has

| Method | # trainable params | Memory cost |
|---|---|---|
| AdamW | $mn$ | $3mn$ |
| GaLore [82] | $mn$ | $mn + mr + 2nr$ |
| POET (FS) | $b(b-1)$ | $mn + 3b(b-1)$ |
| POET (BS) | $\frac{1}{2}(m+n)(b-1)$ | $mn + \frac{3}{2}(m+n)(b-1)$ |

Table 1: Comparison to existing methods. Assume $W \in \mathbb{R}^{m \times n}$ ($m \leq n$), GaLore with rank $r$ and POET with block size $b$. FS denotes fully stochastic SPO, and BS denotes block-stochastic SPO.

parameter complexity dependent on the matrix size (*i.e.*, $m + n$), making it more scalable than AdamW, which requires $mn$ trainable parameters. In terms of memory complexity, both POET variants can be much more efficient than AdamW with a suitable sampling budget $b$. A comparison of parameter and memory complexity is given in Table 1.

**Performance under a constant parameter budget**. Since POET optimizes two orthogonal matrices $\boldsymbol{R}, \boldsymbol{P}$ simultaneously, a natural question arises: *which matrix should receive more parameter budget under a fixed total constraint?* To investigate this, we conduct a controlled experiment where different ratios of trainable parameters are allocated to $\boldsymbol{R}$ and $\boldsymbol{P}$ under a fixed total budget. All other settings (*e.g.*, architecture, data) remain unchanged, with full details provided in the Appendix. We use validation perplexity as the

Figure 5: Performance of POET under a constant total parameter budget on $\boldsymbol{R}, \boldsymbol{P}$.

evaluation metric. The total parameter budget matches that of fully stochastic POET with $b = \frac{1}{h}m$ for $\boldsymbol{R}$ and $b = \frac{1}{h}n$ for $\boldsymbol{P}$, where $h = 8, 4$, and $3$ correspond to small, medium, and large budgets, respectively. We explore seven allocation settings: $\boldsymbol{R} : \boldsymbol{P} = 1 : 0$ (*i.e.*, orthogonal training [48, 51, 65]), $0.9 : 0.1$, $0.75 : 0.25$, $0.5 : 0.5$ (*i.e.*, standard POET), $0.25 : 0.75$, $0.1 : 0.9$, and $0 : 1$. Results in Figure 5 show that POET with a balanced allocation between $\boldsymbol{R}$ and $\boldsymbol{P}$ yields the best performance.

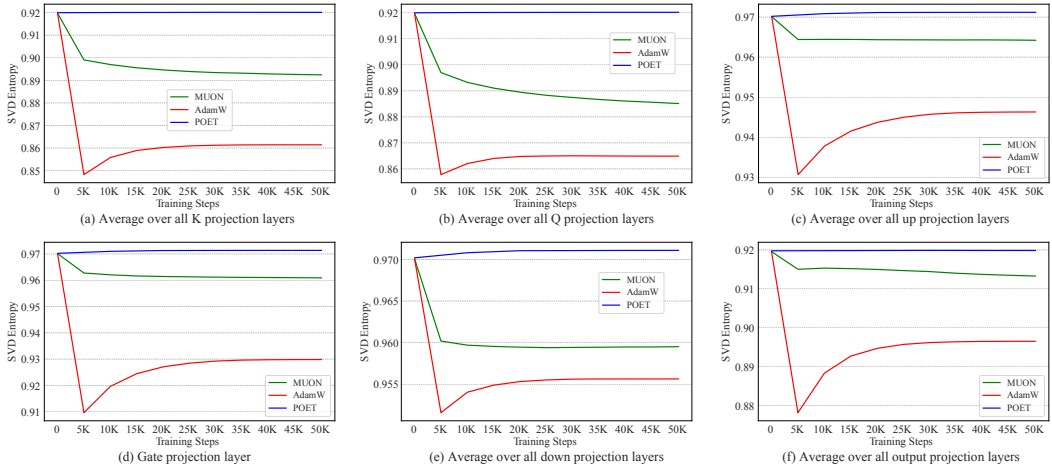

Figure 6: Dynamics comparison of average singular value entropy (singular value diversity) between direct training (AdamW, Muon) and POET.

**Guarantees of weight spectrum**. For POET with standard and normalized Gaussian initializations, we have proved in Appendix C that the largest and smallest singular values of weights can be bounded. For normalized Gaussian, the bound is only dependent on the row-to-column ratio of the weight matrix. For standard Gaussian, the bound has an extra dependence on the neuron dimension.

**Connection to generalization theory**. Several generalization results [6, 63, 76] based on bounding the spectral norm of weight matrices. In particular, the spectrally-normalized margin analysis in [6] bounds the misclassification error in terms of a margin-based training loss and a complexity term. The complexity term is proportional to $Q/(\gamma n)$ where $\gamma$ and $n$ are margin and sample size and $Q$ bounds the spectral complexity. For an $L$-layer ReLU MLP and maximal width $d$, $Q$ is bounded by

$$Q = \left( \prod_{i=1}^{L} \|\boldsymbol{W_i}\| \right) \left( \sum_{i=1}^{L} \frac{(\sqrt{d}\|\boldsymbol{W_i}\|_F)^{2/3}}{\|\boldsymbol{W_i}\|^{2/3}} \right)^{3/2} \tag{7}$$

where $\|\cdot\|$ and $\|\cdot\|_F$ denote spectral and Frobenius norm respectively. Those norms remain invariant when training the network with POET and at initialization they can be bounded with high probability using standard results from random matrix theory (Appendix C). The scale at initialization is typically chosen such that $\boldsymbol{W} \in \mathbb{R}^{d \times d}$ satisfies $\|\boldsymbol{W}\| = O(1)$ and $\|\boldsymbol{W}\| = O(\sqrt{d})$ so that $Q = O_L(d)$.

**Approximation properties of SPO**. We have seen in Theorem 1 that the factorization $\boldsymbol{RWP}$ with orthogonal matrices $\boldsymbol{R}$ and $\boldsymbol{P}$ is the most general spectrum preserving transformation of $\boldsymbol{W}$. Here we express $\boldsymbol{R}$ and $\boldsymbol{P}$ as products of stochastic primitives, but as we state next, this does not reduce representation power when using sufficiently many primitives.

**Lemma 1.** *If $c \geq \alpha m \ln(m)(m/b)^2$ for some $\alpha > 0$ then with probability at least $1 - m^{-(\alpha-2)}$ over the randomness of the index sets $\boldsymbol{S}^i$ we can express any orthogonal matrix $\boldsymbol{R}$ as a product of $c$ primitives $\boldsymbol{G}_i$ as in Eq. (4). Moreover, the orthogonal matrix $\boldsymbol{G}_i$ depends only on the sets $\boldsymbol{S}^j$ and matrices $\boldsymbol{G}^j$ selected in earlier steps.*

The proof of this lemma can be found in Appendix D. The result extends to Block-stochastic SPO as this is strictly more expressive than fully stochastic SPO. The key idea of the proof is similar to the factorization of orthogonal matrices into a product of Givens rotations. Indeed, by multiplying $\boldsymbol{R}^\top$ with properly chosen primitive matrices $\boldsymbol{G}_i$ we can create zeros below the diagonal for one column after another. Note that each $\boldsymbol{G}_i$ has $b(b-1)/2$ parameters while $\boldsymbol{R}$ has $m(m-1)/2$ parameters, which implies that generally at least $\Omega((m/b)^2)$ primitives are necessary. In Appendix D we also provide a heuristic that with high probability for $c = O(\ln(m)(m/b)^2)$ every orthogonal matrix can be written as a product of $c$ orthogonal primitives $\boldsymbol{G}_i$.

**Inductive bias**. POET-reparameterized neurons result in neural networks that maintain identical architecture and parameter count during inference as conventionally trained networks. While standard training could technically learn equivalent parameters, they consistently fail to do so in practice. This indicates POET provides a unique inductive bias unavailable through standard training. POET also aligns with prior findings in [2, 18] that optimizing factorized matrices yields implicit inductive bias.

**Dynamics of singular spectrum**. Inspired by [46], we conduct a spectral analysis by comparing the singular spectrum among AdamW, Muon and POET. Following [1, 68], we compute SVD

entropy of the trained Llama-60M model at different iteration. Specifically, the SVD entropy, defined as $H(\boldsymbol{\sigma}) = \frac{1}{\log n} \sum_i \frac{\sigma_i^2}{\sum_j \sigma_j^2} \log \frac{\sigma_i^2}{\sum_j \sigma_j^2}$ measures the diversity of singular values; higher entropy indicates a more uniform and diverse spectrum. [46] attributes the favorable performance of Muon over AdamW to the more diverse spectrum of weight matrices updates. As shown in Figure 6, POET consistently maintains high spectral diversity throughout training, owing to its orthogonal equivalence transformation. Therefore, POET can better explore diverse optimization directions.

**POET minimizes hyperspherical energy**. While POET generalizes energy-preserving training to spectrum-preserving training, it still ensures low hyperspherical energy when initialized with a zero-mean isotropic Gaussian distribution. This is significant, as POET retains the generalization benefits of minimal hyperspherical energy [47–49]. This property comes from the invariance of zero-mean isotropic Gaussian to orthogonal transformation. [48] shows that for a weight matrix $\boldsymbol{W}$ initialized by zero-mean isotropic Gaussian, its neurons, after being normalized, are uniformly distributed on the unit hypersphere. This property provably leads to a small hyperspherical energy. Since zero-mean isotropic Gaussian is invariant to orthogonal transformation, $\text{OET}(\boldsymbol{W}; \boldsymbol{R}, \boldsymbol{P}) = \boldsymbol{R}\boldsymbol{W}\boldsymbol{P}$ does not change the distribution of $\boldsymbol{W}$, *i.e.*, $\boldsymbol{R}\boldsymbol{W}\boldsymbol{P} =_d \boldsymbol{W}$. We give a derivation in Appendix B. Therefore, the transformed neurons of the new weight matrix $\boldsymbol{R}\boldsymbol{W}\boldsymbol{P}$ are also uniformly distributed over the unit hypersphere after being normalized. This validates that POET provably preserves a small energy.

The property that POET preserves singular spectrum while retaining small hyperspherical energy well justifies why POET yields stable training and good generalization. Such a property only holds true when the weight initialization follows zero-mean isotropic Gaussian, which is perfectly satisfied by current weight initialization. It may also justify why zero-mean isotropic Gaussian initialization works better than the uniform spectrum initialization in our experiments. To further validate that POET maintains a small hyperspherical energy, we plot the hyperspherical energy of the Llama-60M model

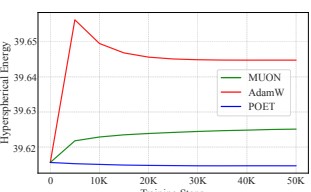

Figure 7: Hyperspherical energy comparison between AdamW, Muon and POET.

at different iterations in Figure 7. We compare the sum of the hyperspherical energy of all the layers trained by AdamW, Muon or POET. The results again verify POET's energy-minimizing property. Different from previous orthogonal training [48], POET can not preserve exactly the same hyperspherical energy during training, but it can well minimize hyperspherical energy in an expectation sense. We also find that Muon can better minimize hyperspherical energy than AdamW.

## 5 Experiments and Results

We start by evaluating POET on large-scale LLaMA pretraining, followed by an extensive ablation study to justify our design choices. Detailed settings and additional results are given in Appendices.

### 5.1 LLM Pretraining using LLaMA Transformers

We perform the pretraining experiments on the Llama transformers of varying sizes (60M, 130M, 350M, 1.3B) for POET. We use the C4 dataset [66], a cleaned web crawl corpus from Common Crawl, widely used for LLM pretraining [29, 56, 82]. For POET-BS, $b$ is the block size of the block-diagonal orthogonal matrix. For POET-FS, $b_{\text{in}}=bm$ for $\boldsymbol{R}$ and $b_{\text{out}}=bn$

| Model (# tokens) | 60M (30B) | 130M (40B) | 350M (40B) | 1.3B (50B) |
|---|---|---|---|---|
| AdamW | 26.68 (25.30M) | 20.82 (84.93M) | 16.78 (302.38M) | 14.73 (1.21B) |
| Galore | 29.81 (25.30M) | 22.35 (84.93M) | 17.99 (302.38M) | 18.33 (1.21B) |
| LoRA$_{r=64}$ | 39.70 (4.85M) | 32.07 (11.21M) | 25.19 (30.28M) | 20.55 (59.38M) |
| POET$_{\text{BS},b=64}$ | 29.52 (2.39M) | 24.52 (5.52M) | 20.29 (14.90M) | 18.28 (29.22M) |
| POET$_{\text{BS},b=128}$ | 26.90 (4.81M) | 21.86 (11.12M) | 18.05 (30.04M) | 16.24 (58.91M) |
| POET$_{\text{BS},b=256}$ | **25.29** (9.66M) | **19.88** (22.33M) | 16.27 (60.32M) | 14.56 (118.26M) |
| POET$_{\text{FS},b=1/8}$ | 34.06 (0.53M) | 29.67 (1.78M) | 24.61 (6.34M) | 18.46 (25.39M) |
| POET$_{\text{FS},b=1/4}$ | 28.69 (2.13M) | 23.55 (7.13M) | 19.42 (25.44M) | 17.60 (101.66M) |
| POET$_{\text{FS},b=1/2}$ | 25.37 (8.54M) | 19.94 (28.56M) | **15.95** (101.86M) | **13.70** (406.88M) |

Table 2: Comparison of POET with popular pretraining methods using different sizes of LLaMA models. Validation perplexity and the number of trainable parameters are reported.

for $\boldsymbol{P}$. We compare POET against GaLore [82], a low-rank pretraining method, and AdamW, the standard pretraining optimizer. We generally follow the settings in [82]. To better simulate the practical pretraining setting, we significantly increase the number of training tokens for all methods.

Table 2 shows that both POET-FS ($b=1/2$) and POET-BS ($b=256$) consistently outperform both GaLore and AdamW with significantly fewer parameters. For LLaMA-1B, POET-FS ($b=1/2$) yields the best overall performance, achieving a validation perplexity of 13.70, much better than AdamW (14.73) and GaLore (18.33). Block-stochastic POET with $b=256$ achieves the second-best performance

(14.56), which still surpasses AdamW with only one-tenth of AdamW's trainable parameters. Similar patterns can be observed for models of smaller sizes. Moreover, we compare the training dynamics between AdamW and POET in Figure 8. The training dynamics of POET is quite different

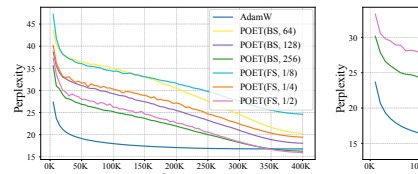 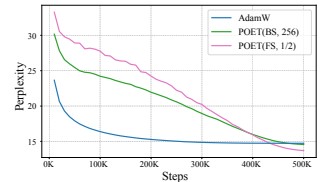

Figure 8: Validation perplexity dynamics on LLaMA-350M and LLaMA-1.3B.

from AdamW. After an initial rapid drop in perplexity, POET improves more slowly than AdamW. As seen in Phase II (Figure 2), this slower but stable progress can lead to better performance in later stages. We attribute this intriguing phenomenon to the unique reparameterization of POET and How we efficiently approximate orthogonality. The exact mechanism behind this phenomenon remains an open question, and understanding it could offer valuable insights into large-scale model training.

To highlight POET's non-trivial performance improvement, we increase the training steps (*i.e.*, effectively tokens seen) for AdamW, and find that POET-FS ($b$=1/2) still outperforms AdamW even even if AdamW is trained with almost triple the number of tokens. Results are given in Figure 9. In this experiment, the AdamW learning rate was carefully tuned for the full training run, and no training tokens were repeated. Thus, the improvement is non-trivial and cannot be attributed to merely increasing training steps. Interestingly, we also observe from Table 2 that POET's performance appears strongly correlated with the parameter budget and larger budgets consistently yield better results across model scales. This is particu-

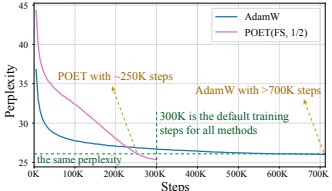

Figure 9: Validation perplexity dynamics of POET (FS, $b$=1/2) and AdamW on Llama-60M. POET outperforms the AdamW trained with almost twice the number of seen tokens.

larly important for model scaling law [35]. Another notable observation is that POET significantly outperforms LoRA [26] given a similar parameter budget. For instance, with approximately 30M trainable parameters, POET attains a validation perplexity of 18.05, significantly better than LoRA's 25.19. We further observe that the block-stochastic variant is more parameter-efficient than the fully stochastic one. On the 130M model, it achieves a validation perplexity of 19.88 with nearly 6M fewer trainable parameters, compared to 19.94 for the fully stochastic variant. We hypothesize that this is due to better coverage of weight parameters. Specifically, the block-stochastic variant ensures all corresponding weights are updated at each step, unlike the more uneven updates in the fully stochastic variant. Experimental details and results on weight update coverage are provided in Appendix H.

**Pretraining Llama-3B**. To demonstrate the scalability of POET, we apply POET to pretrain Llama with 3B parameters while reusing the same hyperparameters from the 1.3B model. This model was trained with only 1/10 of the tokens used for the 1.3B model. Compared to Table 2, the slightly higher final perplexity is attributed to the reduced training data (5B tokens). Table 3 shows that POET maintains the performance advantage over AdamW at the 3B scale,

| Method | Perplexity |
|---|---|
| AdamW | 19.61 |
| POET$_{FS,b=1/2}$ | **16.90** |

Table 3: Validation perplexity of training a LLaMA 3B model on 5 billion tokens.

consistent with the conclusion drawn from Table 2 for smaller models. Our findings confirm that the benefits of POET are not limited to smaller models but can extend robustly to larger scales.

## 5.2 LLM Finetuning on Downstream Tasks

To better evaluate models beyond the validation perplexity, we show the results of finetuning the trained model on the GLUE benchmark [75]. This benchmark provides a comprehensive assessment of a model's

| FT | Model | CoLA | MNLI | MRPC | QNLI | QQP | RTE | SST-2 | STS-B |
|---|---|---|---|---|---|---|---|---|---|
| Full FT | AdamW | 0.361 | 0.658 | 0.696 | 0.818 | 0.829 | 0.534 | 0.914 | **0.880** |
| | POET | **0.523** | **0.818** | **0.824** | **0.885** | **0.902** | **0.661** | **0.920** | 0.873 |
| OFT | AdamW | 0.388 | 0.774 | 0.689 | 0.842 | 0.867 | 0.531 | 0.915 | 0.762 |
| | POET | **0.437** | **0.812** | **0.740** | **0.855** | **0.877** | **0.538** | **0.924** | **0.791** |
| POET | AdamW | 0.435 | 0.804 | 0.806 | 0.856 | 0.889 | 0.653 | 0.904 | 0.878 |
| | POET | **0.505** | **0.821** | **0.826** | **0.892** | **0.902** | **0.682** | **0.931** | **0.887** |

Table 4: Downstream performance on GLUE between AdamW- and POET-pretrained models.

language understanding capabilities through a diverse set of downstream tasks. The performance on GLUE serves as an important indicator of the transferability of the model's learned representations. Performance was measured using accuracy for MNLI, MRPC, QNLI, QQP, RTE, and SST-2; the Matthews Correlation Coefficient for CoLA; and the Pearson Correlation Coefficient for STS-B. For each task, we finetune the models for 10 epochs using a consistent learning rate of 2e-5. The evaluation covers several finetuning strategies: full finetuning (Full FT) with AdamW, orthogonal finetuning (OFT) [51, 65], and finetuning with POET. The results in Table 4 show that the POET-pretrained

model consistently outperforms the AdamW-pretrained baseline across all tasks and finetuning methods. The results further validate POET's generalizability. In this setting, OFT uses significantly less parameters than both Full FT and POET, so OFT generally performs worse. However, the comparison between Full FT and POET is fair, since both methods update the entire model.

### 5.3 Ablation Studies and Empirical Analyses

**Initialization schemes**. We empirically compare different random initialization schemes for POET, including two commonly used ones (standard Gaussian, Xavier [17]) and two proposed ones (uniform spectrum, normalized Gaussian). Specifically, we use fully stochastic POET with $b=1/2$ to train Llama-60M on 30B tokens and report the validation perplexity in Table 5. Results show that the normalized initialization will lead to the best final performance, and we stick to it as a default choice. Interestingly, uniform

| Scheme | Perplexity |
|---|---|
| Standard | 26.22 |
| Xavier | 25.79 |
| Uni. spectrum | 27.29 |
| Normalized | **25.37** |

Table 5: Performance of different initializations.

spectrum initialization performs poorly. This suggests a trade-off between preserving good weight spectral properties and achieving strong expressiveness. it may limit its expressiveness. Finding the optimal singular value structure for weights remains an important open problem.

**Merge-then-reinitialize frequency**. The proposed merge-then-reinitialize trick allows POET to train only a small fraction of the large orthogonal matrices $R$ and $P$ per iteration, significantly reducing GPU memory usage. However, this trick also introduces a reinitialization frequency hyperparameter $T_m$, which determines how often the orthogonal matrix is merged and reset to the identity. The index set in POET-FS and the permutation matrix in POET-BS are also resampled at each reinitialization. Therefore, it is quite important to understand how this hyperparameter $T_m$ affects performance. Following the previous initialization experiment, we

| $T_m$ | Perplexity |
|---|---|
| 5 | 30.29 |
| 25 | 27.27 |
| 50 | 25.99 |
| 200 | 25.37 |
| 400 | **25.31** |
| 1600 | 25.58 |

Table 6: Val. perplexity of different $T_m$.

use POET-FS with $b=1/2$ to train Llama-60M on 30B tokens. We vary the reinitialization frequency from 5 to 1600 and report the validation perplexity in Table 6. Results show that both 200 and 400 perform well. Therefore, we set $T_m = 400$ in all experiments by default.

**Neumann series approximation**. CNP approximates the matrix inverse using a Neumann series. As the number of Neumann terms directly influences the approximation quality, understanding its impact on model performance is essential. To this end, we evaluate how varying the number of Neumann terms affects performance, using POET-FS with $b = 1/2$ to train LLaMA-130M. Results in Table 7 show that increasing the number of Neumann terms generally improves validation perplexity. However, this also leads to slower training.

| Scheme | Perplexity |
|---|---|
| $k = 0$ | Not converged |
| $k = 1$ | 22.56 |
| $k = 2$ | 21.54 |
| $k = 3$ | 20.22 |
| $k = 4$ | **20.19** |

Table 7: Number of terms in Neumann series.

Moreover, dropping all Neumann terms ($k = 0$) leads to training divergence, highlighting the critical role of maintaining orthogonality. To balance overhead and performance, we find that 3 Neumann terms ($k = 3$) reach a good trade-off.

Additionally, we evaluate the accuracy of the Neumann approximation to understand how the number of Neumann terms affects the orthogonality. The orthogonal approximation error is defined by $e_{\text{orth}} = \|RR^T - I\|_F / \|I\|_F$. We randomly

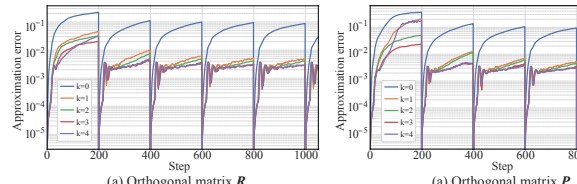

(a) Orthogonal matrix $R$      (a) Orthogonal matrix $P$

Figure 10: Approximation error of orthogonal matrices $R$ and $P$ of a weight matrix.

select a weight matrix and compute the approximation error of corresponding orthogonal matrices $R$ and $P$. For clarity, we visualize the error in the initial 1000 training steps in Figure 10. We observe that, with more Neumann terms, the orthogonal approximation error is indeed lower. Moreover, the merge-then-reinitialize trick can periodically reset the error. More results are given in Appendix L

## 6 Related Work, Concluding Remarks and Acknowledgement

**Related work**. Inspired by low-rank adaptation methods such as LoRA [26], a number of recent approaches [11, 20, 27, 30–32, 43–45, 53, 58, 71, 81, 82] have explored low-rank structures to enable efficient pretraining of large language models (LLMs). In parallel, sparsity has also been extensively studied as a means to improve training efficiency in neural networks [9, 14, 15, 25, 72, 78]. Compared to approaches that exploit low-rank structures, relatively few works have explored sparsity for pretraining. Our work broadly aligns with the sparse training paradigm, as POET leverages

sparsely optimized orthogonal matrices to enhance training efficiency. A parallel line of research [34, 46, 60, 69, 80] focuses on developing efficient optimizers for large-scale neural networks. While our work also targets efficient training of large models, it is orthogonal to these efforts, as POET can be integrated with any optimizer. The way POET uses orthogonal matrices to transform neurons may also relate to preconditioned optimizers such as Muon [34], Shampoo [19] and SOAP [73], as well as to the broader field of manifold optimization (e.g., [7]). POET-trained weight matrices remain statistically indistinguishable from randomly initialized ones due to the isotropy of zero-mean independent Gaussian distributions. This yields interesting connections to random neural networks [21, 40, 40, 62, 74], random geometry [22], and random matrix theory [16].

**Concluding remarks**. This paper introduces POET, a reparameterized training algorithm for large language models. POET models each neuron as the product of two orthogonal matrices and a fixed random weight matrix. By efficiently learning large orthogonal transformations, POET achieves superior generalization while being much more parameter-efficient than existing LLM pretraining methods. Experiments show that POET is broadly applicable to both pretraining and finetuning tasks.

**Acknowledgement**. The authors would like to sincerely thank Lixin Liu, Han Shi, Gege Gao, Zhen Liu and many colleagues at Max Planck Institute for Intelligent Systems for many helpful suggestions. Additionally, the authors also sincerely thank all the anonymous NeurIPS reviewers for their constructive suggestions that greatly improved the quality of our work.

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

# Appendix

## Table of Contents

# A  Delving into POET's Three Training Phases

## A.1  More Details on Vector Probing

The three training phases of POET are summarized from the empirical observation of the vector probing results. The idea of vector probing is very straightforward. We generate a constant vector $v$ that is randomly initialized. Then we let it to be transformed by the learned orthogonal matrices $R$ and $P$. Finally, we compute the cosine of their angle: $v^\top R v$ and $v^\top P v$. In this process, the probing vector $v$ is always fixed. The full results are given in Appendix G.

Beyond a particular constant probing vector, we also consider a set of randomly sampled probing vectors that follow our proposed normalized Gaussian initialization. Specifically, we consider the following expectation:

$$\mathbb{E}_{v \sim \mathbb{S}^{m-1}}\{v^\top R v\}, \tag{8}$$

where $v$ is a vector initialized by normalized Gaussian distribution (thus uniformly distributed on a unit hypersphere $\mathbb{S}^{m-1}$). Because $\mathbb{E}\{vv^\top\} = \frac{1}{m}$, then we have that

$$\mathbb{E}_{v \sim \mathbb{S}^{m-1}}\{v^\top R v\} = \frac{1}{m}\mathrm{Tr}(R). \tag{9}$$

where $\mathrm{Tr}(\cdot)$ denotes the matrix trace. Its geometric interpretation is the cosine of the rotation angle between $v$ and $Rv$.

Next, we look into the variance of $q(x) = v^\top R v$ (we simplify the expectation over the unit hypersphere to $\mathbb{E}$):

$$\mathrm{Var}(q(x)) = \mathbb{E}\{(v^\top R v)^2\} - (\mathbb{E}\{v^\top R v\})^2. \tag{10}$$

First we compute $\mathbb{E}\{(v^\top R v)^2\}$:

$$
\begin{aligned}
\mathbb{E}\{(v^\top R v)^2\} &= \frac{\mathrm{Tr}(R)^2 + 2\left\|\frac{R^\top + R}{2}\right\|}{m(m+2)} \\
&= \frac{\mathrm{Tr}(R)^2 + \mathrm{Tr}(R^2) + m}{m(m+2)}
\end{aligned}
\tag{11}
$$

Then we compute $(\mathbb{E}\{v^\top R v\})^2$:

$$(\mathbb{E}\{v^\top R v\})^2 = \frac{\mathrm{Tr}(R)^2}{m^2}. \tag{12}$$

Finally, we combine pieces and have the final variance:

$$\mathrm{Var}(v^\top R v) = \frac{m + \mathrm{Tr}(R^2) + \frac{2\mathrm{Tr}(R)^2}{m}}{m(m+2)} \tag{13}$$

which shrinks at the order of $O(1/m)$. Therefore, when the dimension of orthogonal matrices is large, even if we use a fixed random probing vector $v$, this rotation angle is quite consistent.

## A.2  Geometric Interpretation of the Trace of Orthogonal Matrices

Let's delve deeper into the trace of orthogonal matrices. It generally represents how much a transformation preserves vectors in their original directions. Specifically, the trace indicates how much "alignment" or similarity there is between the original vectors and their images after transformation.

The trace of an orthogonal matrix $R \in \mathbb{R}^{m \times m}$ can be written as

$$\mathrm{Tr}(R) = \sum_{i=1}^{m} e_i^\top R e_i \tag{14}$$

where $e_i, \forall i$ are unit basis vectors. This expression reveals that the trace measures the sum of inner products between each original direction $e_i$ and its transformed version $Re_i$. Since $e_i^\top R e_i$ can be interpreted as the cosine of the angle between $e_i$ and $Re_i$, the trace thus reflects how much the orthogonal transformation aligns with or deviates from the original coordinate directions.

We also plot the trace of both $\boldsymbol{R}$ and $\boldsymbol{P}$ during the POET training. The results are shown in Figure 13 and Figure 14. After dividing the trace by the orthogonal matrix dimension, we obtain that the result is generally in the range of $[0.6, 0.65]$ after training. This is similar to the results of vector probing. Therefore, we empirically verify the conclusion that the expectation of vector probing results is $\frac{\text{Tr}(\boldsymbol{R})}{m}$ with a small variance.

## A.3 Empirical Observations

The training dynamics of POET presents three geometry-driven phases. We note that these phase changes are based on empirical observation, and further theoretical understanding of this process remains an open problem.

**Phase I: conical-shell searching** rotates each orthogonal matrix $R$ and $P$ smoothly away from the identity while preserving their singular values, so the cosine similarity between transformed and initial weight vectors falls from 1 to $\approx 0.6$; this provides a spectrally well-conditioned "cone" in which learning can proceed safely. this phase serves the role of "spectral warm-up". By plotting the cosine similarity of any one layer, we always see the same graceful slide towards 0.6–0.65, independent of model size, layer type, or whether you train with fully-stochastic or block-stochastic SPO. This phase carves out the thin "shell" in which subsequent learning lives.

**Phase II: stable learning on the conical shell** occupies the bulk of training: the angles to the initial vectors stay locked in that narrow band, optimization now shears weights *within* the cone, and validation perplexity drops almost linearly because spectra remain frozen and gradients act only on meaningful directions. In this phase, the trace of the orthogonal matrices stay almost as a constant.

Specifically, we hypothesize that the orthogonal transforms have reached a "good" cone; thereafter they mostly shear vectors inside that shell, leaving the angle to the original vector unchanged. The spectrum continues to be exactly that of the random initial matrix, so gradients can no longer distort singular values and instead devote capacity to learning meaningful directions. Because the geometry is stabilized in this phase, the learning of patterns happen in a stable subspace. This stable learning phase takes up 80% of the training time.

**Phase III: final adjusting** coincides with learning-rate decay; the orthogonal transforms barely move, making only tiny refinements to singular vectors, so additional steps yield diminishing returns. This phase is merely the LR cooldown; weights and spectra are already near their final configuration, so progress naturally slows.

# B    Minimum Hyperspherical Energy in POET

We start by showing that orthogonal equivalence transformation in POET can provably obtain small hyperspherical energy for its transformed weight matrices. This result holds when the weight matrices are independently initialized by zero-mean isotropic Gaussian distribution. Both Xavier [17] and Kaiming [23] initializations satisfy such a weight initialization condition. Orthogonal equivalence transformation is given by $\text{OET}(\boldsymbol{W}; \boldsymbol{R}, \boldsymbol{P}) = \boldsymbol{R}\boldsymbol{W}\boldsymbol{P}$, where the input matrix $\boldsymbol{W}$ is multiplied on the left and on the right by orthogonal matrices $\boldsymbol{R}$ and $\boldsymbol{P}$, respectively.

When $\boldsymbol{W}$ is initialized by zero-mean isotropic Gaussian distribution, [48] has shown that these random neurons, if normalized, are uniformly distributed on the unit hypersphere. This leads to a provably small hyperspherical energy for the randomly initialized weight matrix. In the following, we will show that after orthogonal equivalence transformation, the weight matrix still maintains a small hyperspherical energy.

We consider a weight matrix $\boldsymbol{W} \in \mathbb{R}^{m \times n}$ where each entry is *i.i.d.* sampled from a zero-mean Gaussian distribution with variance the same variance $\sigma^2$, *i.e.*, $W_{ij} \sim \mathcal{N}(0, \sigma^2)$. After applying orthogonal equivalence transformation, we have $\boldsymbol{W}^{\text{new}} = \boldsymbol{R}\boldsymbol{W}\boldsymbol{P}$ where $\boldsymbol{R}$ and $\boldsymbol{P}$ are two orthogonal matrices. Then we compute the distribution of $\boldsymbol{W}^{\text{new}}$. Because linear maps preserve Gaussianity, each entry of $\boldsymbol{W}^{\text{new}}$ is a finite linear combination of $W_{ij}$, and hence, $\boldsymbol{W}^{\text{new}}$ follows a joint Gaussian which can be fully characterized by mean and covariance.

The mean of $\boldsymbol{W}^{\text{new}}$ is given by

$$\mathbb{E}[\boldsymbol{W}^{\text{new}}] = \boldsymbol{R} \cdot \mathbb{E}[\boldsymbol{W}] \cdot \boldsymbol{P} = \boldsymbol{R} \cdot \boldsymbol{0} \cdot \boldsymbol{P} = \boldsymbol{0} = \mathbb{E}[\boldsymbol{W}]. \tag{15}$$

For its covariance, we consider two generic entries $W_{ij}^{\text{new}}$ and $W_{i'j'}^{\text{new}}$:

$$
\begin{aligned}
W_{ij}^{\text{new}} &= \sum_{k,l} R_{ik} W_{kl}^{\text{new}} P_{lj}, \\
W_{i'j'}^{\text{new}} &= \sum_{u,v} R_{i'u} W_{uv}^{\text{new}} P_{vj'}.
\end{aligned}
\tag{16}
$$

Then we compute the covariance between the two entries:

$$
\begin{aligned}
\text{Cov}(W_{ij}^{\text{new}}, W_{i'j'}^{\text{new}}) &= \sum_{k,l,u,v} R_{ik} R_{i'u} P_{lj} P_{vj'} \text{Cov}(W_{kl}, W_{uv}) \\
&= \sigma^2 (\boldsymbol{R}\boldsymbol{R}^\top)_{ii'} (\boldsymbol{P}^\top \boldsymbol{P})_{jj'} \\
&= \sigma^2 \delta_{ii'} \delta_{jj'}
\end{aligned}
\tag{17}
$$

which implies the following resutls:

- The covariance matrix is a diagonal matrix, so different entries of $\boldsymbol{W}^{\text{new}}$ are uncorrelated.
- Because $\boldsymbol{W}^{\text{new}}$ is a joint Gaussian and different entries are uncorrelated, each entry of $\boldsymbol{W}^{\text{new}}$ is independent.
- Each entry of $\boldsymbol{W}^{\text{new}}$ has identical variance $\sigma^2$.

To sum up, each entry of $\boldsymbol{W}^{\text{new}}$ is *i.i.d.* $\mathcal{N}(0, \sigma^2)$, which is identical to the distribution of each entry of $\boldsymbol{W}$. Because we have $\boldsymbol{W}^{\text{new}} =_d \boldsymbol{W}$, we can conclude that, similar to $\boldsymbol{W}$, $\boldsymbol{W}_{\text{new}}$ also has provably small hyperspherical energy among neurons.

Despite being extremely simple, we find that this is in fact a significant result. Under zero-mean isotropic Gaussian initialization, spectrum-preserving training and energy-preserving training can be achieved simultaneously. It also partially explains why the proposed normalized Gaussian initialization achieves the best performance.

# C    Guarantees of Weight Spectrum under POET

For standard Gaussian initialization where each element of the weight matrix $\boldsymbol{W} \in d \times n$ is sampled with a normal distribution, we have the following standard results [8, 70]:

$$\frac{1}{\sqrt{d}}\sigma_{\max}(\boldsymbol{W}) \xrightarrow[n\to\infty]{\text{a.s.}} 1 + \sqrt{\lambda}$$

$$\frac{1}{\sqrt{d}}\sigma_{\min}(\boldsymbol{W}) \xrightarrow[n\to\infty]{\text{a.s.}} 1 - \sqrt{\lambda} \tag{18}$$

which gives spectrum guarantees for weight matrices generated by the standard Gaussian initialization.

In the following, we give the spectrum guarantees for the normalized Gaussian initialization. We start by stating the following theorem from [49]:

**Theorem 2.** *Let $\tilde{\boldsymbol{v}}_1, \cdots, \tilde{\boldsymbol{v}}_n \in \mathbb{R}^d$ be i.i.d. random vectors where each element follows the Gaussian distribution with mean $0$ and variance $1$. Then $\boldsymbol{v}_1 = \frac{\tilde{\boldsymbol{v}}_1}{\|\tilde{\boldsymbol{v}}_1\|_2}, \cdots, \boldsymbol{v}_n = \frac{\tilde{\boldsymbol{v}}_n}{\|\tilde{\boldsymbol{v}}_n\|_2}$ are uniformly distributed on the unit hypersphere $\mathbb{S}^{d-1}$. If the ratio $\frac{n}{d}$ converges to a constant $\lambda \in (0,1)$, asymptotically we have for $\boldsymbol{W} = \{\boldsymbol{v}_1, \cdots, \boldsymbol{v}_n\} \in \mathbb{R}^{d \times n}$:*

$$\lim_{n\to\infty} \sigma_{\max}(\boldsymbol{W}) \leq (\sqrt{d} + \sqrt{\lambda d}) \cdot (\max_i \frac{1}{\|\tilde{\boldsymbol{v}}_i\|_2})$$

$$\lim_{n\to\infty} \sigma_{\min}(\boldsymbol{W}) \geq (\sqrt{d} - \sqrt{\lambda d}) \cdot (\min_i \frac{1}{\|\tilde{\boldsymbol{v}}_i\|_2}) \tag{19}$$

*where $\sigma_{\max}(\cdot)$ and $\sigma_{\min}(\cdot)$ denote the largest and the smallest singular value of a matrix, respectively.*

*Proof.* We first introduce the following lemma as the characterization of a unit vector that is uniformly distributed on the unit hypersphere $\mathbb{S}^{d-1}$.

**Lemma 2** ([64]). *Let $\boldsymbol{v}$ be a random vector that is uniformly distributed on the unit hypersphere $\mathbb{S}^{d-1}$. Then $\boldsymbol{v}$ has the same distribution as the following:*

$$\left\{ \frac{u_1}{\sqrt{\sum_{i=1}^d u_i^2}}, \frac{u_2}{\sqrt{\sum_{i=1}^d u_i^2}}, \cdots, \frac{u_d}{\sqrt{\sum_{i=1}^d u_i^2}} \right\} \tag{20}$$

*where $u_1, u_2, \cdots, u_d$ are i.i.d. standard normal random variables.*

*Proof.* The lemma follows naturally from the fact that the Gaussian vector $\{u_i\}_{i=1}^d$ is rotationally invariant. □

Then we consider a random matrix $\tilde{\boldsymbol{W}} = \{\tilde{\boldsymbol{v}}_1, \cdots, \tilde{\boldsymbol{v}}_n\}$ where $\tilde{\boldsymbol{v}}_i$ follows the same distribution of $\{u_1, \cdots, u_d\}$. Therefore, it is also equivalent to a random matrix with each element distributed normally. For such a matrix $\tilde{\boldsymbol{W}}$, we have from [70] that

$$\lim_{n\to\infty} \sigma_{\max}(\tilde{\boldsymbol{W}}) = \sqrt{d} + \sqrt{\lambda d}$$

$$\lim_{n\to\infty} \sigma_{\min}(\tilde{\boldsymbol{W}}) = \sqrt{d} - \sqrt{\lambda d} \tag{21}$$

where $\sigma_{\max}(\cdot)$ and $\sigma_{\min}(\cdot)$ denote the largest and the smallest singular value, respectively.

Then we write the matrix $\boldsymbol{W}$ as follows:

$$\boldsymbol{W} = \tilde{\boldsymbol{W}} \cdot \boldsymbol{Q}$$

$$= \tilde{\boldsymbol{W}} \cdot \begin{bmatrix} \frac{1}{\|\tilde{\boldsymbol{v}}_1\|_2} & 0 & \cdots & 0 \\ 0 & \frac{1}{\|\tilde{\boldsymbol{v}}_2\|_2} & \ddots & 0 \\ \vdots & \ddots & \ddots & \vdots \\ 0 & \cdots & 0 & \frac{1}{\|\tilde{\boldsymbol{v}}_n\|_2} \end{bmatrix} \tag{22}$$

which leads to

$$\lim_{n\to\infty} \sigma_{\max}(\boldsymbol{W}) = \lim_{n\to\infty} \sigma_{\max}(\tilde{\boldsymbol{W}} \cdot \boldsymbol{Q})$$
$$\lim_{n\to\infty} \sigma_{\min}(\boldsymbol{W}) = \lim_{n\to\infty} \sigma_{\min}(\tilde{\boldsymbol{W}} \cdot \boldsymbol{Q})$$

(23)

We fist assume that for a symmetric matrix $\boldsymbol{A} \in \mathbb{R}^{n\times n}$ $\lambda_1(\boldsymbol{A}) \geq \cdots \geq \lambda_n(\boldsymbol{A})$. Then we introduce the following inequalities for eigenvalues:

**Lemma 3** ([57]). *Let* $\boldsymbol{G}, \boldsymbol{H} \in \mathbb{R}^{n\times n}$ *be positive semi-definite symmetric, and let* $1 \leq i_1 < \cdots < i_k \leq n$. *Then we have that*

$$\prod_{t=1}^{k} \lambda_{i_t}(\boldsymbol{G}\boldsymbol{H}) \leq \prod_{t=1}^{k} \lambda_{i_t}(\boldsymbol{G})\lambda_t(\boldsymbol{H})$$

(24)

*and*

$$\prod_{t=1}^{k} \lambda_{i_t}(\boldsymbol{G}\boldsymbol{H}) \geq \prod_{t=1}^{k} \lambda_{i_t}(\boldsymbol{G})\lambda_{n-t+1}(\boldsymbol{H})$$

(25)

*where* $\lambda_i$ *denotes the* $i$-*th largest eigenvalue.*

We first let $1 \leq i_1 < \cdots < i_k \leq n$. Because $\tilde{\boldsymbol{W}} \in \mathbb{R}^{d\times n}$ and $\boldsymbol{Q} \in \mathbb{R}^{n\times n}$, we have the following:

$$\prod_{t=1}^{k} \sigma_{i_t}(\tilde{\boldsymbol{W}}\boldsymbol{Q}) = \prod_{t=1}^{k} \sqrt{\lambda_{i_t}(\tilde{\boldsymbol{W}}\boldsymbol{Q}\boldsymbol{Q}^\top\tilde{\boldsymbol{W}}^\top)}$$
$$= \sqrt{\prod_{t=1}^{k} \lambda_{i_t}(\tilde{\boldsymbol{W}}^\top\tilde{\boldsymbol{W}}\boldsymbol{Q}\boldsymbol{Q}^\top)}$$

(26)

by applying Lemma 3 to the above equation, we have that

$$\sqrt{\prod_{t=1}^{k} \lambda_{i_t}(\tilde{\boldsymbol{W}}^\top\tilde{\boldsymbol{W}}\boldsymbol{Q}\boldsymbol{Q}^\top)} \geq \sqrt{\prod_{t=1}^{k} \lambda_{i_t}(\tilde{\boldsymbol{W}}^\top\tilde{\boldsymbol{W}})\lambda_{n-t+1}(\boldsymbol{Q}\boldsymbol{Q}^\top)}$$
$$= \prod_{t=1}^{k} \sigma_{i_t}(\tilde{\boldsymbol{W}})\sigma_{n-t+1}(\boldsymbol{Q})$$

(27)

$$\sqrt{\prod_{t=1}^{k} \lambda_{i_t}(\tilde{\boldsymbol{W}}^\top\tilde{\boldsymbol{W}}\boldsymbol{Q}\boldsymbol{Q}^\top)} \leq \sqrt{\prod_{t=1}^{k} \lambda_{i_t}(\tilde{\boldsymbol{W}}^\top\tilde{\boldsymbol{W}})\lambda_t(\boldsymbol{Q}\boldsymbol{Q}^\top)}$$
$$= \prod_{t=1}^{k} \sigma_{i_t}(\tilde{\boldsymbol{W}})\sigma_t(\boldsymbol{Q})$$

(28)

Therefore, we have that

$$\prod_{t=1}^{k} \sigma_{i_t}(\tilde{\boldsymbol{W}}\boldsymbol{Q}) \geq \prod_{t=1}^{k} \sigma_{i_t}(\tilde{\boldsymbol{W}})\sigma_{n-t+1}(\boldsymbol{Q})$$

(29)

$$\prod_{t=1}^{k} \sigma_{i_t}(\tilde{\boldsymbol{W}}\boldsymbol{Q}) \leq \prod_{t=1}^{k} \sigma_{i_t}(\tilde{\boldsymbol{W}})\sigma_t(\boldsymbol{Q})$$

(30)

Suppose we have $k = 1$ and $i_1 = n$, then Eq. (29) gives

$$\sigma_n(\tilde{\boldsymbol{W}}\boldsymbol{Q}) \geq \sigma_n(\tilde{\boldsymbol{W}})\sigma_n(\boldsymbol{Q})$$

(31)

Then suppose we have $k = 1$ and $i_1 = 1$, then Eq. (30) gives

$$\sigma_1(\tilde{\boldsymbol{W}}\boldsymbol{Q}) \leq \sigma_1(\tilde{\boldsymbol{W}})\sigma_1(\boldsymbol{Q})$$

(32)

Combining the above results with Eq. (21) and Eq. (23), we have that

$$
\begin{aligned}
\lim_{n\to\infty} \sigma_{\max}(\boldsymbol{W}) = \lim_{n\to\infty} \sigma_{\max}(\tilde{\boldsymbol{W}} \cdot \boldsymbol{Q}) &\leq \lim_{n\to\infty} \left( \sigma_{\max}(\tilde{\boldsymbol{W}}) \cdot \sigma_{\max}(\boldsymbol{Q}) \right) \\
&= (\sqrt{d} + \sqrt{\lambda d}) \cdot \max_i \frac{1}{\|\tilde{\boldsymbol{v}}_i\|_2} \\
\lim_{n\to\infty} \sigma_{\min}(\boldsymbol{W}) = \lim_{n\to\infty} \sigma_{\min}(\tilde{\boldsymbol{W}} \cdot \boldsymbol{Q}) &\geq \lim_{n\to\infty} \left( \sigma_{\min}(\tilde{\boldsymbol{W}}) \cdot \sigma_{\min}(\boldsymbol{Q}) \right) \\
&= (\sqrt{d} - \sqrt{\lambda d}) \cdot \min_i \frac{1}{\|\tilde{\boldsymbol{v}}_i\|_2}
\end{aligned}
\tag{33}
$$

which concludes the proof. □

Combing with the fact that

$$
\lim_{n\to\infty} \max \frac{\|\boldsymbol{v}_i\|_2}{\sqrt{d}} = \lim_{n\to\infty} \min \frac{\|\boldsymbol{v}_i\|_2}{\sqrt{d}} = 1,
\tag{34}
$$

we essentially have that

$$
\begin{aligned}
\lim_{n\to\infty} \sigma_{\max}(\boldsymbol{W}) &\to 1 + \sqrt{\lambda}, \\
\lim_{n\to\infty} \sigma_{\min}(\boldsymbol{W}) &\to 1 - \sqrt{\lambda}.
\end{aligned}
\tag{35}
$$

which can be written to the following results:

$$
\boxed{
\begin{aligned}
\sigma_{\max}(\boldsymbol{W}) &\xrightarrow[n\to\infty]{\text{a.s.}} 1 + \sqrt{\lambda} \\
\sigma_{\min}(\boldsymbol{W}) &\xrightarrow[n\to\infty]{\text{a.s.}} 1 - \sqrt{\lambda}
\end{aligned}
}
\tag{36}
$$

which shows that under our proposed normalized Gaussian initialization, the maximal and minimal singular values are well bounded by a constant that is only dependent on the size of weight matrix. These results justify the effectiveness of our proposed normalized Gaussian initialization in POET.

# D   Proofs of Lemma 1

*Proof of Lemma 1.* We consider an orthogonal matrix $\boldsymbol{R}$ and orthogonal primitives $\boldsymbol{G}^i$ corresponding to uniformly random subsets $\boldsymbol{S}^j \subset [m]$ of size $b$ as explained in the main text (see equation (4)). The main claim we need to prove is that given any vector $\boldsymbol{v} \in \mathbb{R}^m$ and a set $\boldsymbol{S} \subset [m]$ with $k \in [m]$ we can find an orthogonal primitive matrix $\boldsymbol{G}$ corresponding to the set $\boldsymbol{S}$ such that

$$
\begin{aligned}
(\boldsymbol{Gv})_l &= 0 \quad \text{for } i \in \boldsymbol{S} \text{ with } l > k \\
(\boldsymbol{Gv})_k &\geq 0 \\
(\boldsymbol{Gv})_l &= v_l \quad \text{for } l \notin \boldsymbol{S}.
\end{aligned}
\tag{37}
$$

Moreover, for all $\boldsymbol{w} \in \mathbb{R}^m$ with $\boldsymbol{w}_i = 0$ for $i \geq k$ the relation

$$
\boldsymbol{Gw} = \boldsymbol{w}
\tag{38}
$$

holds. We can assume that the matrix $\boldsymbol{D}(\boldsymbol{S}) = \{\boldsymbol{e}(s_1), \ldots, \boldsymbol{e}(s_b)\}$ contains the entries $s_i$ in ascending order. Then we write

$$
\boldsymbol{D}(\boldsymbol{S})^\top \boldsymbol{v} = \begin{pmatrix} \tilde{\boldsymbol{v}}_1 \\ \tilde{\boldsymbol{v}}_2 \end{pmatrix}
\tag{39}
$$

where $\tilde{\boldsymbol{v}}_1 \in \mathbb{R}^{b_1}$ corresponds to the entries $s_i$ with $s_i < k$ and $\tilde{\boldsymbol{v}}_2 \in \mathbb{R}^{b_2}$ to the remaining entries, in particular $s_{b_1+1} = k$ because $k \in \boldsymbol{S}$. It is well known that for every vector $\boldsymbol{v}$ there is a rotation $\boldsymbol{Q}$ aligning $\boldsymbol{v}$ with the first standard basis vector, i.e., such that $\boldsymbol{Qv} = \lambda \boldsymbol{e}(1)$ for some $\lambda \geq 0$. Consider such a matrix $\tilde{\boldsymbol{Q}}$ for the vector $\tilde{\boldsymbol{v}}_2$ and then define the orthogonal matrix

$$
\tilde{\boldsymbol{G}} = \begin{pmatrix} \boldsymbol{1}_{b_1} & \boldsymbol{0}_{b_1 \times b_2} \\ \boldsymbol{0}_{b_2 \times b_1} & \tilde{\boldsymbol{Q}} \end{pmatrix}.
\tag{40}
$$

Careful inspection of (4) implies that the last part of (37) is actually true for any $\tilde{\boldsymbol{G}}$ as the second term has rows with all entries equal to zero for all $l \notin \boldsymbol{S}$. For the first part we find

$$
\boldsymbol{D}(\boldsymbol{S})\tilde{\boldsymbol{G}}\boldsymbol{D}(\boldsymbol{S})^\top \boldsymbol{v} = \boldsymbol{D}(\boldsymbol{S})\tilde{\boldsymbol{G}}\begin{pmatrix} \tilde{\boldsymbol{v}}_1 \\ \tilde{\boldsymbol{v}}_2 \end{pmatrix} = \boldsymbol{D}(\boldsymbol{S})\begin{pmatrix} \tilde{\boldsymbol{v}}_1 \\ \lambda \boldsymbol{e}(1) \end{pmatrix} = \sum_{i \leq b_1} \boldsymbol{e}(s_i)(\tilde{\boldsymbol{v}}_1)_i + \lambda \boldsymbol{e}(k).
\tag{41}
$$

Here we used $s_{b_1+1} = k$ in the last step. Since in addition

$$
((\boldsymbol{1}_m - \boldsymbol{D}(\boldsymbol{S}) \cdot \boldsymbol{1}_b \cdot \boldsymbol{D}(\boldsymbol{S})^\top)\boldsymbol{v})_l = 0
\tag{42}
$$

for all $l \in \boldsymbol{S}$ we conclude that indeed $(\boldsymbol{Gv})_l = 0$ for $l \in \boldsymbol{S}$ and $l > k$, $(\boldsymbol{Gv})_k \geq 0$. The remaining statement (38) follows from the observation that when decomposing as in (39) we find

$$
(\boldsymbol{D}(\boldsymbol{S}))^\top \boldsymbol{w} = \begin{pmatrix} \tilde{\boldsymbol{w}}_1 \\ \boldsymbol{0}_{b_2} \end{pmatrix}
\tag{43}
$$

(because $\boldsymbol{w}_i = 0$ for $i \geq k$) and therefore

$$
(\tilde{\boldsymbol{G}} - \boldsymbol{1}_b)(\boldsymbol{D}(\boldsymbol{S}))^\top \boldsymbol{w} = \boldsymbol{0}_b
\tag{44}
$$

by definition of $\tilde{\boldsymbol{G}}$ and we find $\boldsymbol{Gw} = \boldsymbol{w}$.

The rest of the proof is straightforward by induction combined with a simple coin collector problem. For the rest of the proof it is convenient to reverse the indices, i.e., to consider products $\boldsymbol{G}_c \cdot \ldots \cdot \boldsymbol{G}_1$ Assume that we have chosen $\boldsymbol{G}_i$ for $i \leq c_k$ and some $c_k \in \mathbb{N}$ such that the product

$$
\boldsymbol{P}^k = \boldsymbol{G}_{c_k} \cdot \ldots \cdot \boldsymbol{G}_1 \cdot \boldsymbol{R}^\top
\tag{45}
$$

satisfies $\boldsymbol{P}^k_{l',k'} = 0$ for all $k' < k$ and $l' > k'$ and $\boldsymbol{P}^k_{k',k'} \geq 0$ for $k' < k$. Let $c_{k+1} \geq c_k + \alpha(m/b)^2 \ln(m)$. Then, we can bound for any $l > k$ the probability that there is no $c_k < j \leq c_{k+1}$ such that $\{k, l\} \subset \boldsymbol{S}^j$ using that $\boldsymbol{S}^j$ follows a uniform i.i.d. distribution by

$$
\mathbb{P}(\nexists c_k < j \leq c_{k+1} : k, l \in \boldsymbol{S}^j) \leq \left(1 - \frac{b^2}{m^2}\right)^{c_{k+1}-c_k} \leq \exp\left(-\frac{b^2}{m^2} \cdot \alpha \frac{m^2}{b^2} \ln(m)\right) = m^{-\alpha}.
\tag{46}
$$

The union bound implies that with probability at least $1 - m^{-\alpha+1}$ there is for all $l > k$ a $c_k < j \leq c_{k+1}$ such that $\{k, l\} \subset \mathbf{S}^j$. If this holds we set $\mathbf{G}_j$ for $c_k < j \leq c_{k_1}$ as constructed above if $k \in \mathbf{S}^j$ and $\mathbf{G}_j = \mathbf{1}_m$ otherwise. This then ensures that

$$\mathbf{P}^{k+1} = \mathbf{G}_{c_{k+1}} \cdot \ldots \cdot \mathbf{G}_1 \cdot \mathbf{R}^\top \tag{47}$$

satisfies $\mathbf{P}^{k+1}_{l',k'} = 0$ for $k' \leq k$ and $l' > k'$. For $k' < k$ this follows from (38) and for $k' = k$ from (37). We conclude by the union bound that $\mathbf{P}^m$ is an upper triangular matrix with non-negative diagonal entries with probability at least $1 - mm^{-\alpha+1} = 1 - m^{-(\alpha-2)}$. But we also know that $\mathbf{P}^m$ is orthogonal and therefore satisfies $\mathbf{P}^m = \mathbf{1}_m$ and we thus find

$$\mathbf{G}_{c_m} \cdot \ldots \cdot \mathbf{G}_1 = \mathbf{R}. \tag{48}$$

$\square$

Next we give a heuristic that actually $O(\ln(m)m^2/b^2)$ terms are sufficient to express every orthogonal map as a product of stochsastic primitives. For fixed $c$ we consider the map

$$\Phi : O(b)^c \to O(m) \quad \Phi(\tilde{\mathbf{G}}_1, \ldots, \tilde{\mathbf{G}}_c) = \prod_{j=1}^c \mathbf{G}_j. \tag{49}$$

If $c \geq \alpha \ln(m)m^2/b^2$ we have that with probability at least $1 - m^{-(\alpha-2)}$ for all $k, l \in [m]$ there is $j \leq c$ such that $k, l \in \mathbf{S}^j$. Assume that this is the case. Recall that the tangent space of $O(k)$ at the identity is the space of skew-symmetric matrices. Consider a tangent vector $(X_1, \ldots, X_c)$ with $X_i \in \mathrm{Skew}(k)$. Then

$$D\Phi(\mathbf{1}_b, \ldots, \mathbf{1}_b)(X_1, \ldots, X_c) = \sum_{j=1}^c \mathbf{D}(\mathbf{S}^j) \cdot X_j \cdot \mathbf{D}(\mathbf{S}^j)^\top. \tag{50}$$

This is a surjective map on $\mathrm{Skew}(m)$ under the condition that for all $k, l \in [m]$ there is $j \leq c$ such that $k, l \in \mathbf{S}^j$. We can therefore conclude that the image of $\Phi$ contains a neighbourhood of the identity. Moreover, since $\Phi$ is a polynomial map, $D\Phi$ is surjective everywhere except for a variety of codimension one. While this is not sufficient to conclude that the image of $\Phi$ is $O(d)$ or dense in $O(d)$ it provides some indication that this is the case.

# E   Experimental Details

| Parameter | Llama 60M | Llama 130M | Llama 350M | Llama 1.3B |
|---|---|---|---|---|
| Hidden dimension | 512 | 768 | 1024 | 2048 |
| Intermediate dimension | 1280 | 2048 | 2816 | 5376 |
| Number of attention heads | 8 | 12 | 16 | 32 |
| Number of hidden layers | 8 | 12 | 24 | 24 |

Table 8: Model architectures for different Llama variants.

| Model | Spec. | # GPU | lr (base) | lr (POET) | training steps | batch size | grad acc. |
|---|---|---|---|---|---|---|---|
| **Llama 60M** | $b = 1/2$ | 1 | 1e-2 | 1e-3 | 300,000 | 256 | 2 |
| | $b = 1/4$ | 1 | 1e-2 | 2e-3 | 300,000 | 256 | 2 |
| | $b = 1/8$ | 1 | 1e-2 | 4e-3 | 300,000 | 256 | 2 |
| **Llama 130M** | $b = 1/2$ | 1 | 5e-3 | 1e-3 | 400,000 | 128 | 2 |
| | $b = 1/4$ | 1 | 5e-3 | 2e-3 | 400,000 | 128 | 2 |
| | $b = 1/8$ | 1 | 5e-3 | 4e-3 | 400,000 | 128 | 2 |
| **Llama 350M** | $b = 1/2$ | 4 | 5e-3 | 1e-3 | 400,000 | 128 | 1 |
| | $b = 1/4$ | 4 | 5e-3 | 2e-3 | 400,000 | 128 | 1 |
| | $b = 1/8$ | 4 | 5e-3 | 4e-3 | 400,000 | 128 | 1 |
| **Llama 1.3B** | $b = 1/2$ | 8 | 1e-3 | 1e-3 | 500,000 | 64 | 1 |
| | $b = 1/4$ | 8 | 1e-3 | 2e-3 | 500,000 | 64 | 1 |
| | $b = 1/8$ | 8 | 1e-3 | 4e-3 | 500,000 | 64 | 1 |

Table 9: Hyper-parameter setup of POET-FS.

This section outlines our experimental setup, including the codebase, datasets, and computational resources used.

**Code framework.**   Our method is implemented on top of the codebase from [82][1] (Apache 2.0 license), which we also use to reproduce the AdamW and GaLore baselines. We will release our code for reproducing all training results prior to publication.

**Training details.**   We employed the AdamW optimizer [55] for all our training runs. The specific hyperparameters used for each experiment are detailed in the Table 9 and Table 10 referenced below. We use the consine learning rate scheduler with the minimum learning ratio of 0.01. We use the number of warmup steps of 0, weight decay of 0.01 and gradient clipping of 0.1. For the AdamW baseline, we report results for the optimal learning rate from $[1\times10^{-2}, 5\times10^{-3}, 1\times10^{-3}, 5\times10^{-4}, 1\times10^{-4}, 5\times10^{-5}, 1\times10^{-5}]$. After each merge-then-reinitialize step, we additionally increase the gradient clipping for 10 training steps to improve training stability.

**Model architecture.**   Our work utilized the **Hugging Face Transformers**[2] code base to construct the Llama model for pretraining, which is under the **Apache 2.0** license. The specific layer setups for the different scaled Llama models are summarized in Table 8. Note, the intermediate dimension of the Feed-Forward Network (FFN) has been slightly modified for the POET-BS, compared to the configs in [82], because the linear layer dimensions have to be divisible by the POET-BS block size $b$.

**Dataset.**   We use the *Colossal Clean Crawled Corpus* (C4) dataset [66] for pretraining. The C4 data is a large-scale, meticulously cleaned version of Common Crawl's web crawl corpus. It was

---

[1]https://github.com/jiaweizzhao/GaLore
[2]https://github.com/huggingface/transformers

| Model | Spec. | # GPU | lr (base) | lr (POET) | training steps | batch size | grad acc. |
|---|---|---|---|---|---|---|---|
| **Llama 60M** | $b = 256$ | 1 | 1e-2 | 1e-3 | 300,000 | 256 | 2 |
| | $b = 128$ | 1 | 1e-2 | 2e-3 | 300,000 | 256 | 2 |
| | $b = 64$ | 1 | 1e-2 | 4e-3 | 300,000 | 256 | 2 |
| **Llama 130M** | $b = 256$ | 1 | 5e-3 | 1e-3 | 400,000 | 256 | 2 |
| | $b = 128$ | 1 | 5e-3 | 2e-3 | 400,000 | 256 | 2 |
| | $b = 64$ | 1 | 5e-3 | 4e-3 | 400,000 | 256 | 2 |
| **Llama 350M** | $b = 256$ | 4 | 5e-3 | 1e-3 | 400,000 | 128 | 1 |
| | $b = 128$ | 4 | 5e-3 | 2e-3 | 400,000 | 128 | 1 |
| | $b = 64$ | 4 | 5e-3 | 4e-3 | 400,000 | 128 | 1 |
| **Llama 1.3B** | $b = 256$ | 8 | 1e-3 | 1e-3 | 500,000 | 64 | 1 |
| | $b = 128$ | 8 | 1e-3 | 2e-3 | 500,000 | 64 | 1 |
| | $b = 64$ | 8 | 1e-3 | 4e-3 | 500,000 | 64 | 1 |

Table 10: Hyper-parameter setup of POET-BS.

originally introduced for training the Text-to-Text Transfer Transformer (T5) model and has since become a standard pre-training dataset for testing training algorithms for pre-training large language models. The dataset is released under the **ODC-BY** license.

**Compute Resources.**   All the training tasks are performed on a **NVIDIA HGX H100 8-GPU System** node with 80GB memory each. Depending on the model scale, we train on 1, 4 or 8 GPUs.

# F Implementation and CUDA Acceleration

To enable efficient POET training, we implement the Cayley–Neumann parameterization. To reduce memory usage, we leverage the structure of the skew-symmetric matrix $Q \in \mathbb{R}^{n \times n}$, where the diagonal entries are zero ($Q_{ii} = 0$) and off-diagonal elements satisfy $Q_{ij} = -Q_{ji}$. This structure allows us to store only the upper triangular part of $Q$ as a vector, reducing the number of trainable parameters from $n^2$ to $n(n-1)/2$. During the forward pass, $Q$ is reconstructed on-the-fly using a specialized CUDA kernel, significantly accelerating this process. In addition, the Neumann approximation removes the need for costly and numerically unstable matrix inversion, offering further computational gains. Overall, training a 1.3B LLaMA model on a single H100 8-GPU node yields a 3.8× speedup over the baseline (*i.e.*, native implementation). Table 11 summarizes the contribution of each component to the overall training time.

| Design | Speed-Up |
|---|---|
| Neumann approximation | 1.5× |
| Skew-symmetric CUDA kernel | 1.3× |
| Total | 3.8× |

Table 11: Method design and clock time speed-up.

# G    Results of Vector Probing for $R$ and $P$

In this ablation study, we perform vector probing on the orthogonal matrices $\boldsymbol{R} \in \mathbb{R}^{m \times m}, \boldsymbol{P} \in \mathbb{R}^{n \times n}$ for all linear layers for all blocks of a 60M Llama model trained with POET-FS. The cosine similarity results are reported in Figure 11 and Figure 12, and the trace results are reported in Figure 13 and Figure 14. Since we want to understand the learning dynamics of the orthogonal matrices, we employ $b = 1$ with POET learning rate of $5 \times 10^{-4}$ to eliminate the need for resampling and reinitialization of the orthogonal matrices. Interestingly, we observe this three-phased learning dynamics across different types of linear layers and different-depth transformer blocks.

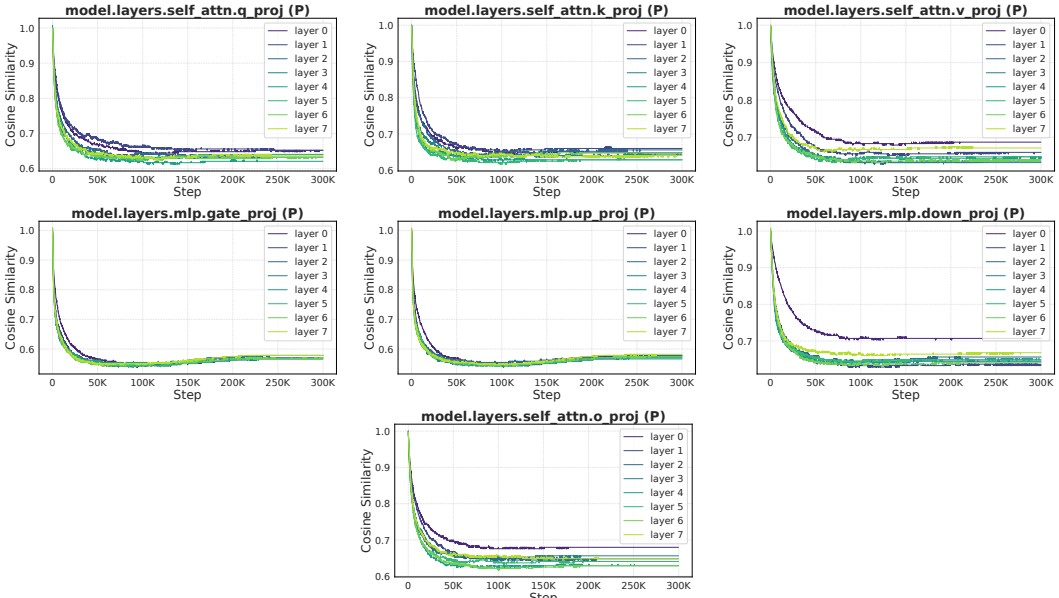

Figure 11: Cosine similarity for vector probing of $\boldsymbol{P}$ across the self-attention components (query, key, value, and output projections) and feed-forward network components (up-, down-, and gate-projections) in all transformer blocks of a POET-trained Llama 60M model.

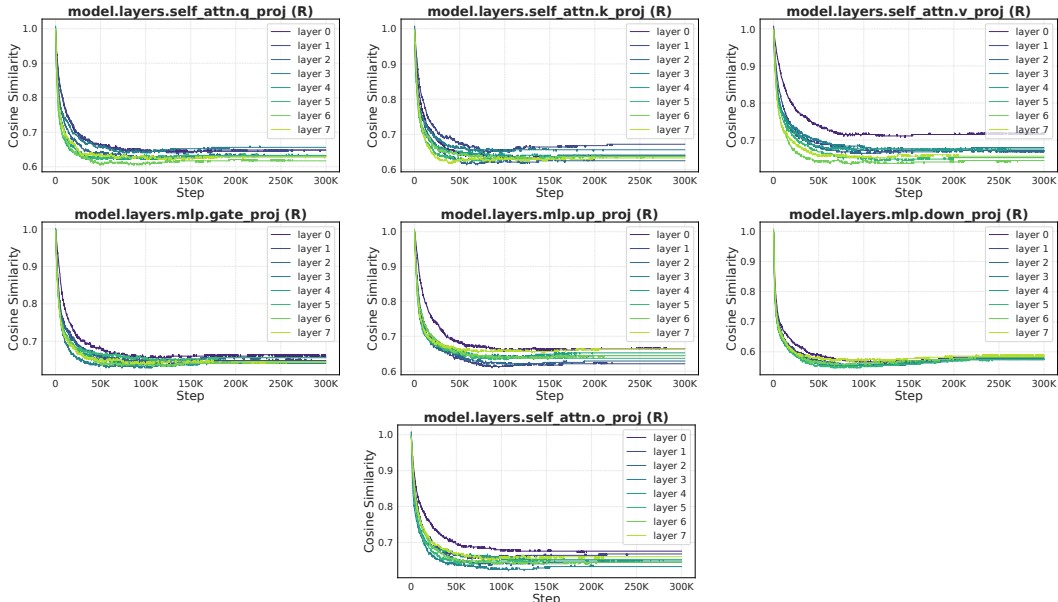

Figure 12: Cosine similarity for vector probing of $\boldsymbol{R}$ across the self-attention components (query, key, value, and output projections) and feed-forward network components (up-, down-, and gate-projections) from all transformer blocks of a POET-trained Llama 60M model.

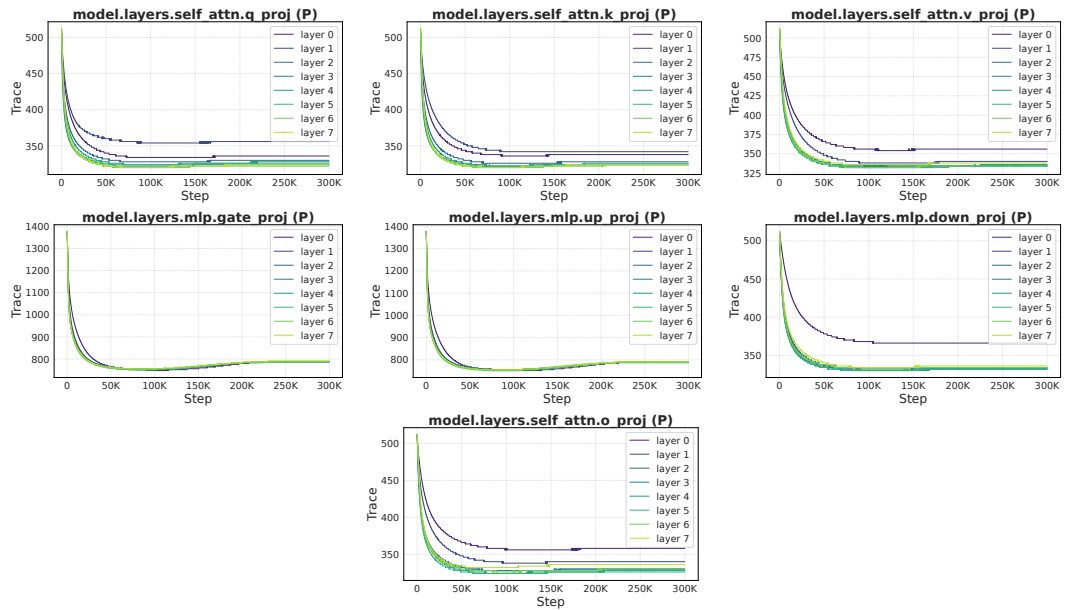

Figure 13: Trace of $\boldsymbol{P}$ across the self-attention components (query, key, value, and output projections) and feed-forward network components (up-, down-, and gate-projections) from all transformer blocks of a POET-trained Llama 60M model.

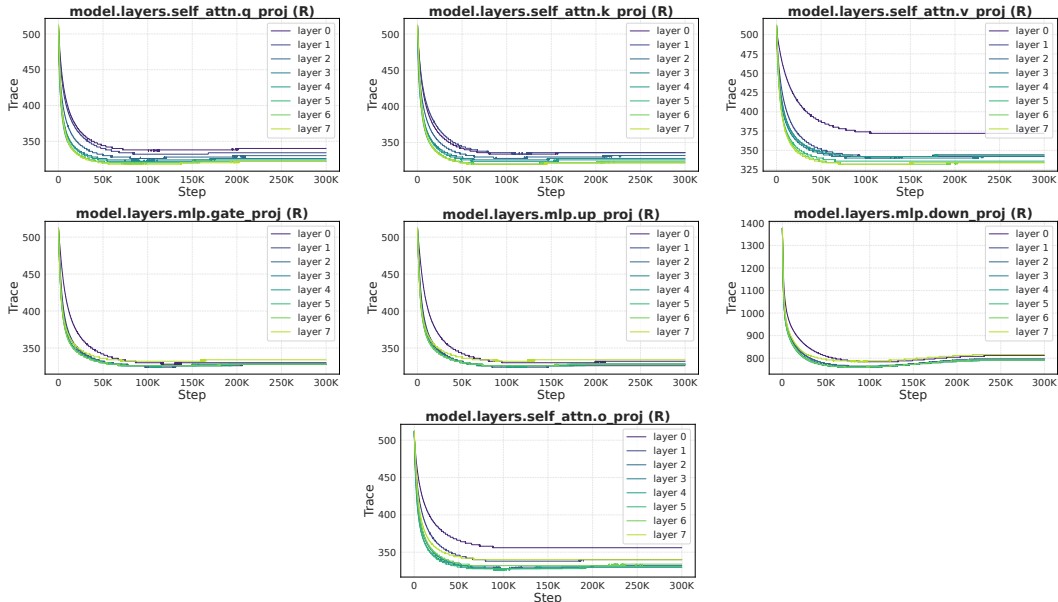

Figure 14: Trace of $\boldsymbol{R}$ across the self-attention components (query, key, value, and output projections) and feed-forward network components (up-, down-, and gate-projections) from all transformer blocks of a POET-trained Llama 60M model.

## H    Weight Update Evenness of Different POET Variants

To understand the higher parameter efficiency of POET-BS compared to POET-FS, we employ a toy example to visualize their different weight update mechanisms by counting the total number of updates for each element of the weight matrix. The visualization results are given in Figure 15 and Figure 16. Specifically, in this experiment, a 64×64 matrix was randomly initialized and trained for 100 steps under various POET-BS and POET-FS configurations. The merge-then-reinitialize trick is performed at each iteration, and the same set of weight elements was effectively updated between two successive merge-then-reinitialize operations. For each weight element, we compute its total number of update in these 100 steps.

Given 100 training steps and updates from both $R$ and $P$, each element of the weight matrix can be updated at most 200 times. This target is consistently achieved by POET-BS, and it is also agnostic to the block size. All POET-BS variants can enable the maximal number of updates for each weight element to be 200. In contrast, POET-FS results in significantly fewer updates per weight element, with updates also unevenly distributed. This unevenness arises from stochasticity, causing certain weights to be updated more frequently than others. While this is less problematic at large iteration counts, it can introduce unexpected training difficulties in earlier stages.

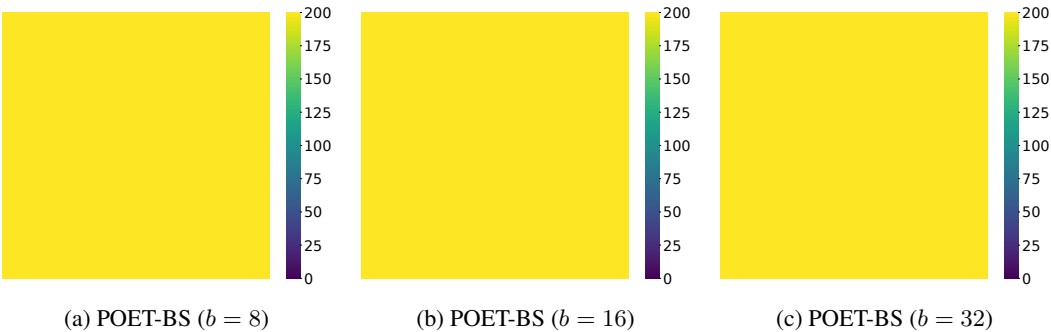

(a) POET-BS ($b = 8$)          (b) POET-BS ($b = 16$)          (c) POET-BS ($b = 32$)

Figure 15: Visualization of the weight update mechanism of POET-BS after 100 steps of update and $T_m = 1$.

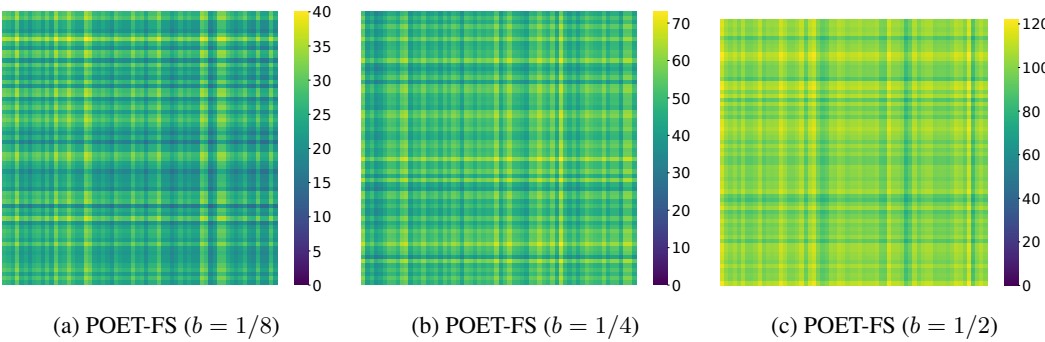

(a) POET-FS ($b = 1/8$)          (b) POET-FS ($b = 1/4$)          (c) POET-FS ($b = 1/2$)

Figure 16: Visualization of the weight update mechanism of POET-FS after 100 steps of update and $T_m = 1$.

# I Training Dynamics of Singular Values

We conduct an ablation study to compare the training dynamics of singular values of weight matrices between **AdamW** and **POET**. The results of AdamW are given in Figure 17, Figure 18 and Figure 19. The results of POET are given in Figure 20, Figure 21 and Figure 22. A 60M LLaMA model was trained for 50,000 iterations with an effective batch size of 512, using both AdamW and POET-FS ($b = 1/2$). The model was evaluated every 5,000 steps, and the singular value dynamics are computed by performing singular value decomposition on the weight matrices. For POET, a merge-then-reinitialize step was applied before each evaluation. Training is finished at 50,000 steps, as the spectral norm of the AdamW-trained model plateaued at this point.

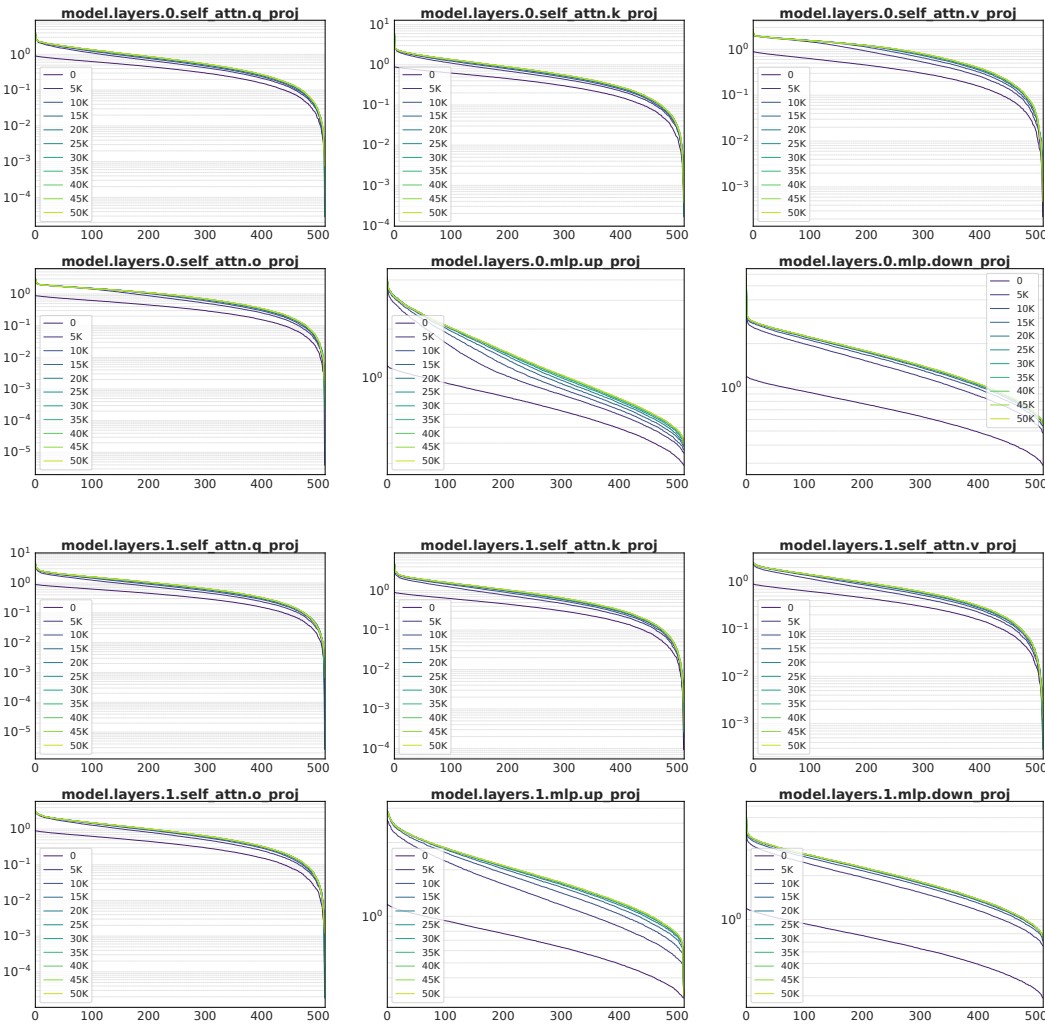

Figure 17: Training dynamics of the singular values of weight matrices within Blocks 0–1 (the $i$-th row represents Block $i$) of a 60M Llama Transformer trained with **AdamW**.

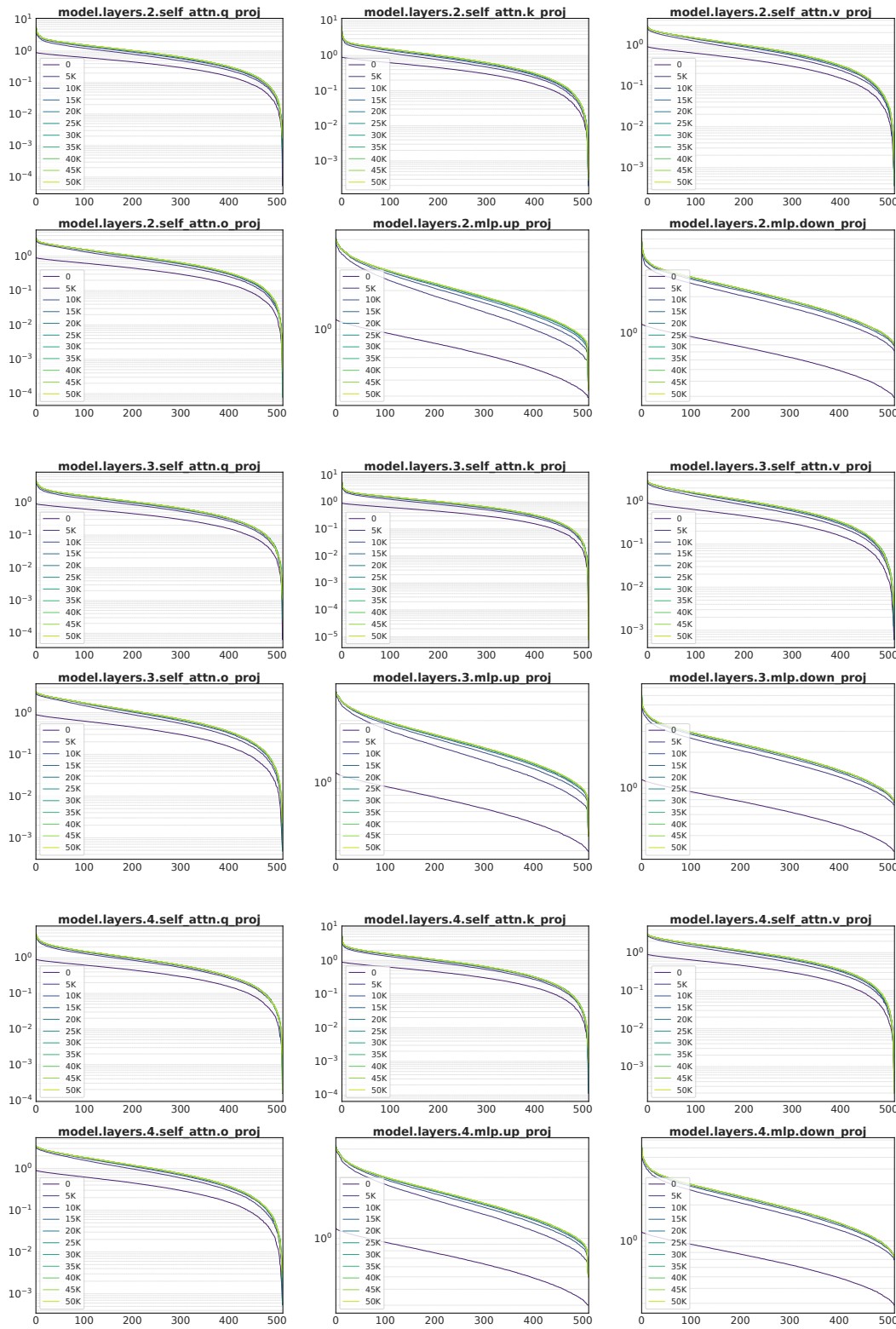

Figure 18: Training dynamics of the singular values of weight matrices within Blocks 2–4 (the $i$-th row represents Block $i$) of a 60M Llama Transformer trained with **AdamW**.

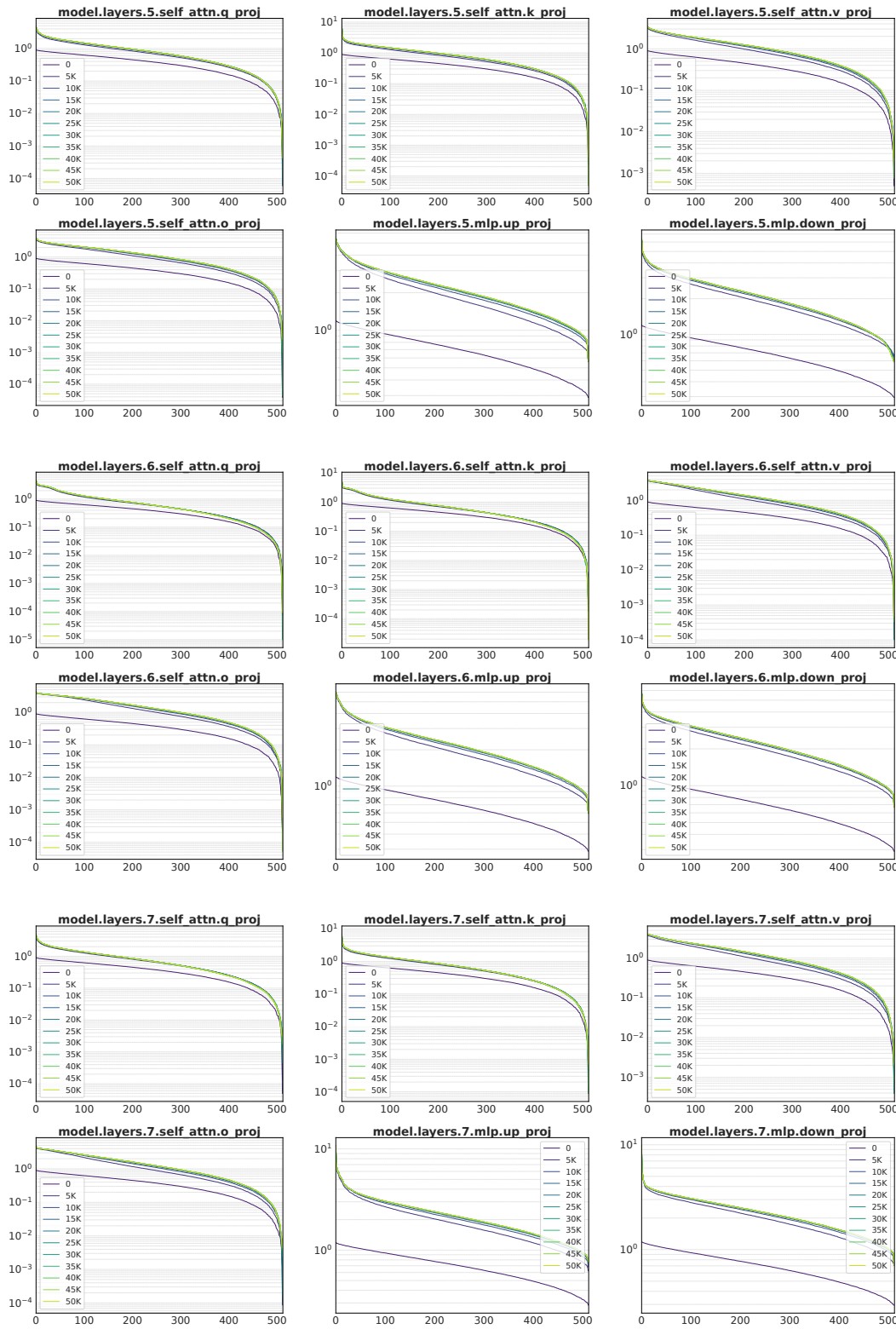

Figure 19: Training dynamics of the singular values of weight matrices within Blocks 5–7 (the $i$-th row represents Block $i$) of a 60M Llama Transformer trained with **AdamW**.

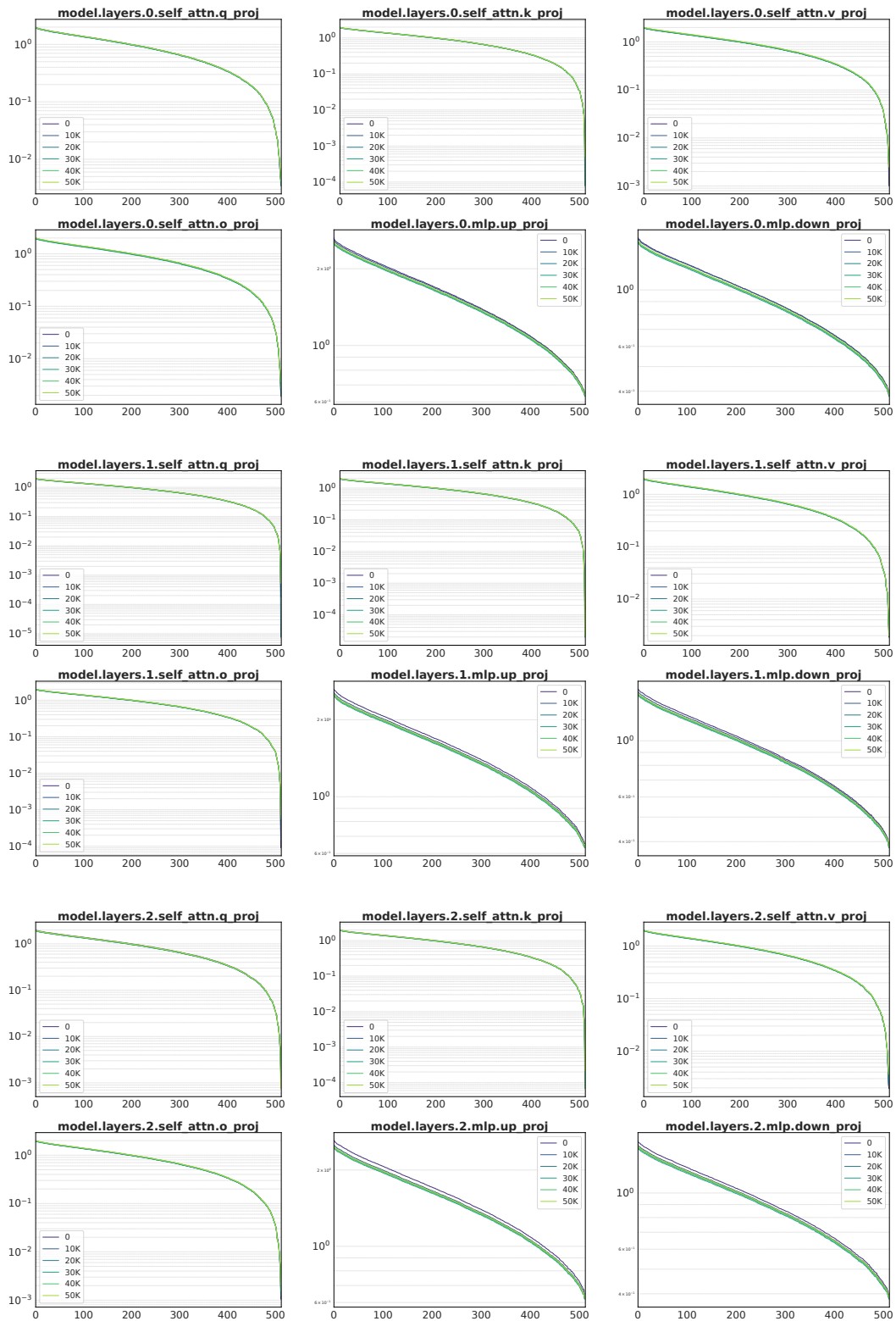

Figure 20: This plot illustrates the singular value training dynamics for individual weight matrices within Blocks 0-2 of a 60M Llama transformer model trained with **POET**. For each block, the dynamics are shown for the self-attention components (query, key, value, and output projections) and the feed-forward network components (up-projection, down-projection, and gate-projection).

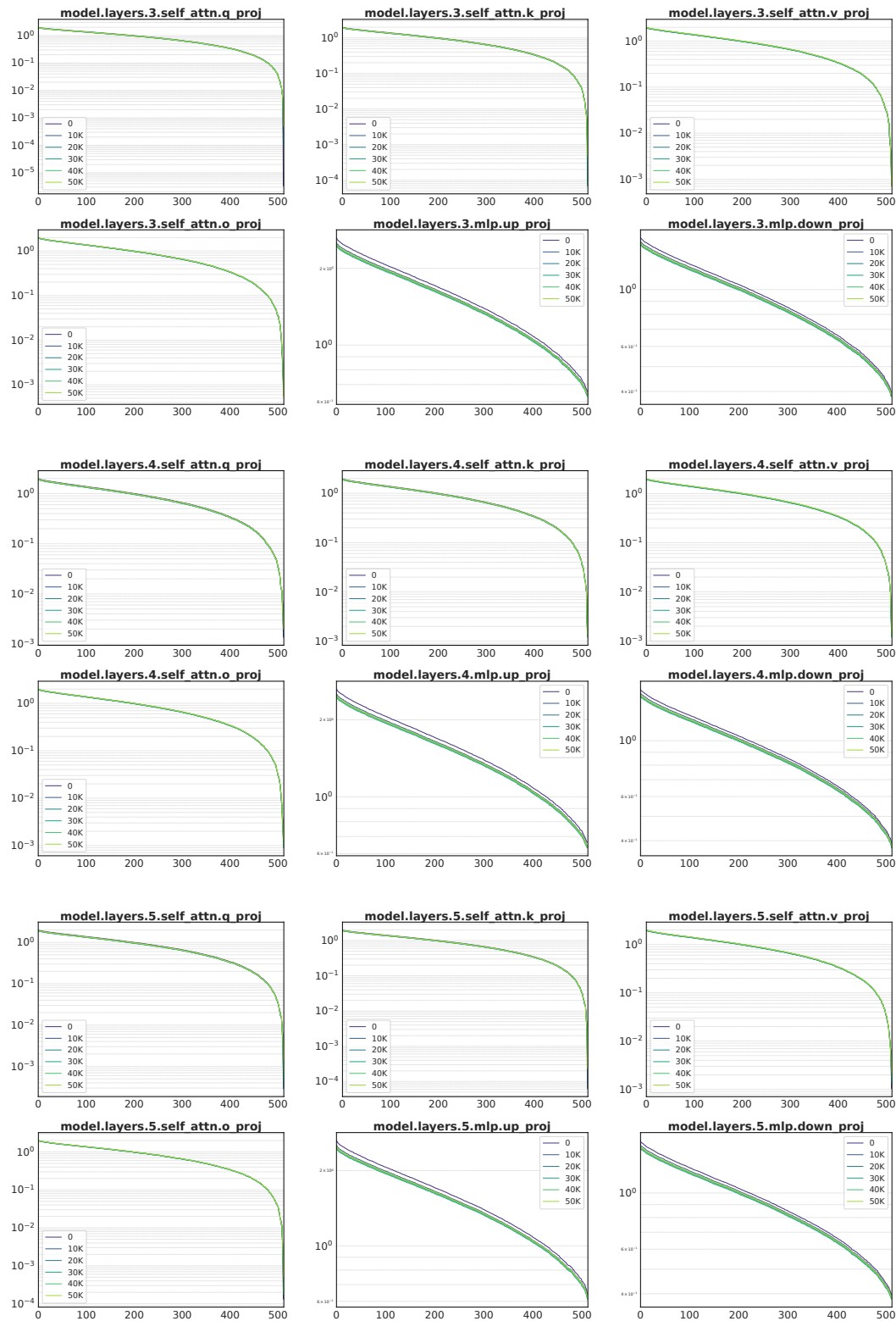

Figure 21: This plot illustrates the singular value training dynamics for individual weight matrices within Blocks 3-5 of a 60M Llama transformer model trained with **POET**. For each block, the dynamics are shown for the self-attention components (query, key, value, and output projections) and the feed-forward network components (up-projection, down-projection, and gate-projection).

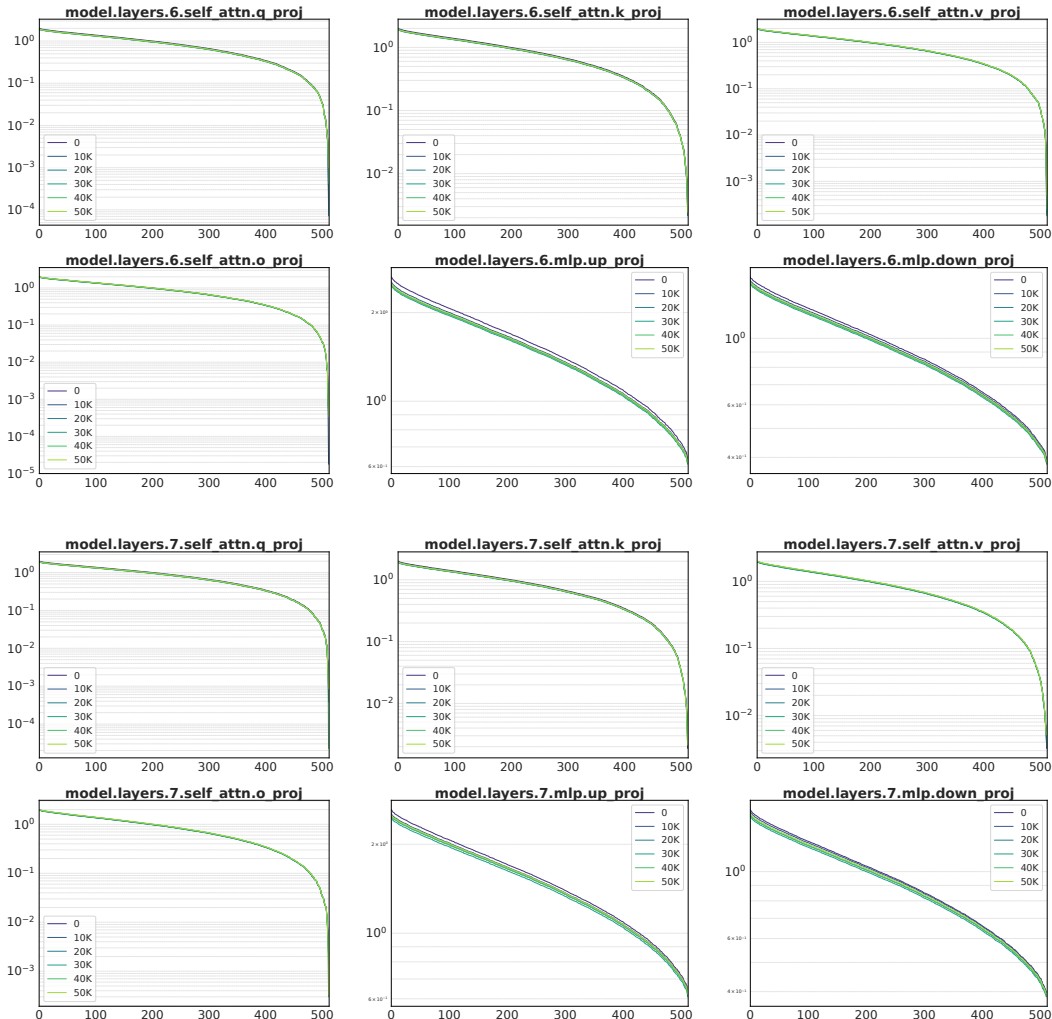

Figure 22: This plot illustrates the singular value training dynamics for individual weight matrices within Blocks 6-7 of a 60M Llama transformer model trained with **POET**. For each block, the dynamics are shown for the self-attention components (query, key, value, and output projections) and the feed-forward network components (up-projection, down-projection, and gate-projection).

# J  Orthogonality Approximation Quality using Neumann Series

In this ablation study, we evaluate the approximation error of the orthogonal matrices $R \in \mathbb{R}^{m \times m}$ and $P \in \mathbb{R}^{n \times n}$ across all linear layers in Block 0 of a 130M LLaMA model trained with POET-FS ($b = 1/2$) for 10,000 steps. Figure 23 and Figure 24 show the approximation error over the first 1,000 steps. Since the error difference between $k = 3$ and $k = 4$ was negligible, we used $k = 3$ for better computational efficiency. Empirically, while $k = 1$ or $k = 2$ suffices for smaller LLaMA models, larger $k$ values are needed to avoid training divergence caused by exploding gradients due to approximation error.

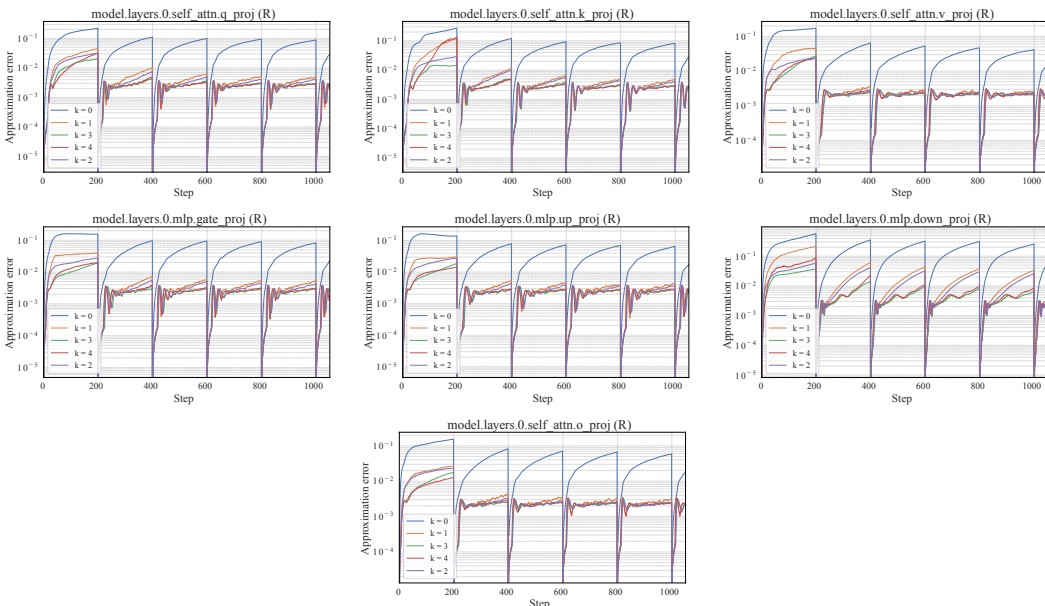

Figure 23: For the transformer block 0, we show approximation error of orthogonal matrix $R$ for the self-attention components (query, key, value, and output projections) and the feed-forward network components (up-projection, down-projection, and gate-projection).

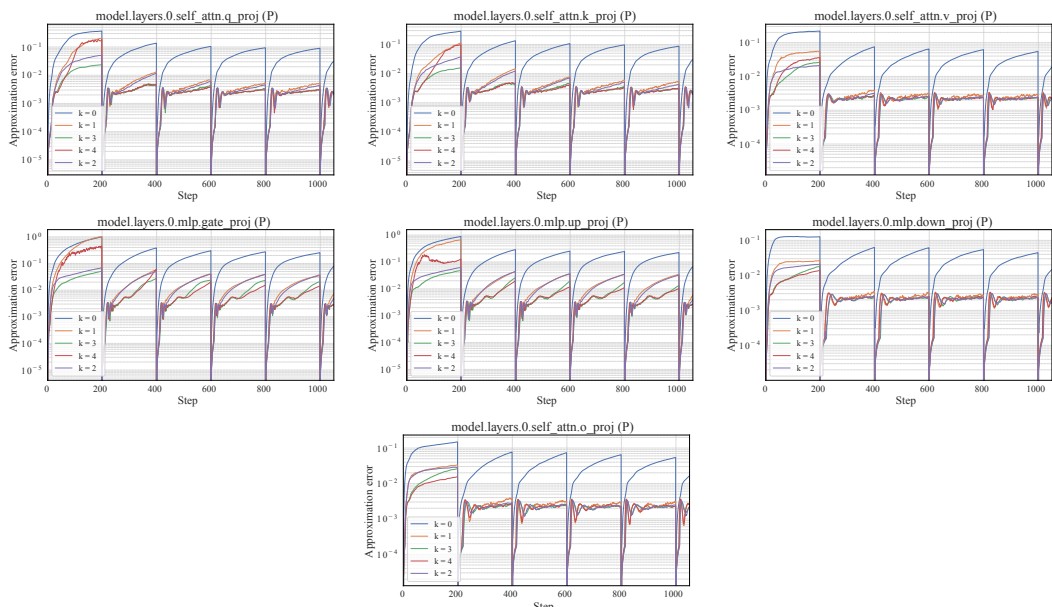

Figure 24: For the transformer block 0, we show approximation error of orthogonal matrix $P$ for the self-attention components (query, key, value, and output projections) and the feed-forward network components (up-projection, down-projection, and gate-projection).

Additionally, Figure 25 shows the orthogonality approximation error of Neumann series with different $k$ over the first 10,000 training steps, illustrating how it decreases as training progresses. We observe a general downward trend in approximation error, indicating improved approximation over time. The results also suggest that using too few Neumann series terms can lead to training divergence in POET.

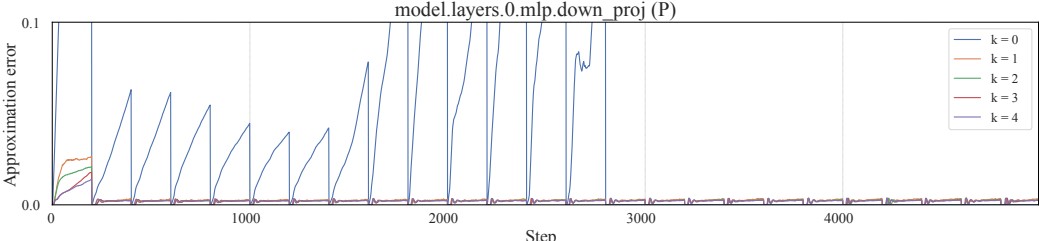

Figure 25: The approximation error of orthogonal matrix $P$ in a randomly selected down-projection layer after training 10000 steps.

# K    Full Results of Training Dynamics

We provide the full training dynamics of different POET variants under Llama 60M, Llama 130M, Llama 350M and Llama 1.3B in Figure 26. This figure is essentially an extended result of Figure 8. One can observe that the training dynamics of POET is quite different from AdamW, and more importantly, POET consistently yields better parameter-efficiency and generalization.

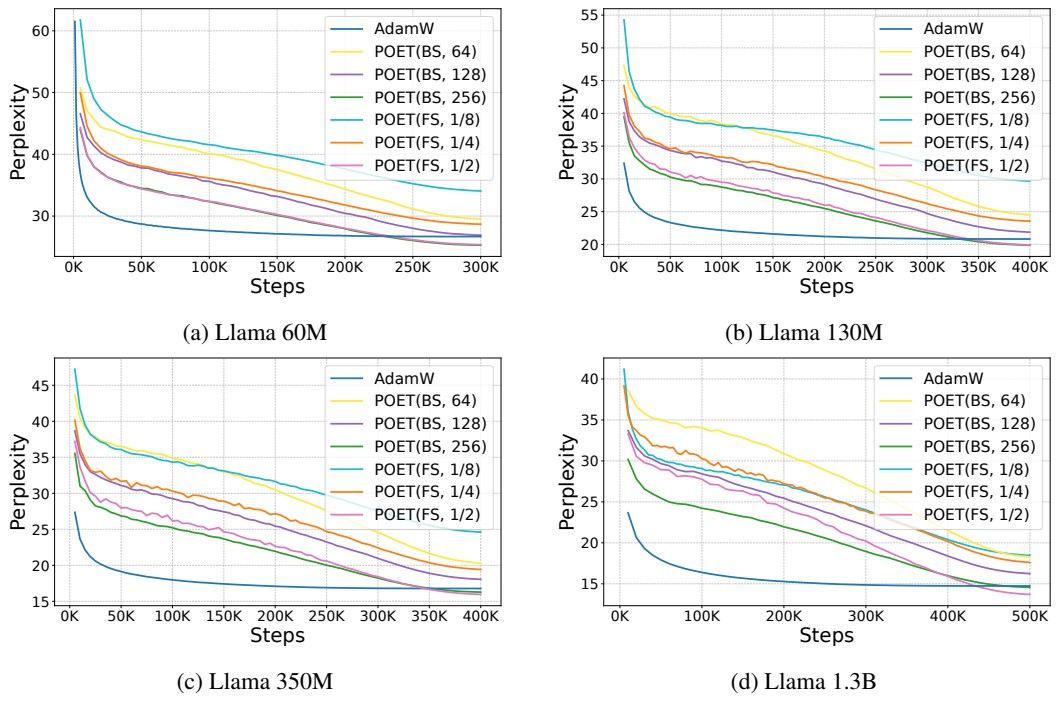

(a) Llama 60M

(b) Llama 130M

(c) Llama 350M

(d) Llama 1.3B

Figure 26: Validation perplexity during the training of the LLama-based transformer with 60M, 130M, 350M and 1.3B parameters.

# L  More Results of POET as a Finetuning Method

To demonstrate the applicability of POET to general finetuning tasks, we apply it to finetune a BART-large model [41] on the NLP task of text summarization. Specifically, we evaluate POET on the XSum [61] and CNN/DailyMail [24] datasets, reporting ROUGE-1/2/L scores in Table 12. We note that both LoRA and OFT are designed solely for parameter-efficient finetuning and are not applicable to pretraining. Our goal here is to demonstrate that POET is also effective as a finetuning method. For consistency, we use the same configuration as in the pretraining setup, resulting in a higher parameter count. Experimental results show that POET not only supports finetuning effectively but also outperforms both full-model finetuning and parameter-efficient methods.

| Method | # Params | XSum | CNN/DailyMail |
|---|---|---|---|
| LoRA ($r$=32) | 17.30M | 43.38 / 20.20 / 35.25 | 43.17 / 20.31 / 29.72 |
| OFT ($b$=64) | 8.52M | 44.12 / 20.96 / 36.01 | 44.08 / 21.02 / 30.68 |
| Full FT | 406.29M | 45.14 / 22.27 / 37.25 | 44.16 / 21.28 / 40.90 |
| POET (FS,$b$=1/2) | 144.57M | **45.23 / 22.41 / 37.28** | **44.27 / 21.29 / 41.02** |

Table 12: Finetuning BART-large on XSum and CNN/DailyMail for text summarization. We report ROUGE-1/2/L results (higher is better).

