# OpenReview forum: "Reparameterized LLM Training via Orthogonal Equivalence Transformation"
_NeurIPS.cc/2025/Conference — NeurIPS 2025 poster_

### Official Review · Reviewer_jtaX · 2025-07-04

**Clarity:** 3
**Significance:** 3
**Originality:** 3
**Rating:** 5
**Confidence:** 3

**Summary:**

This paper proposes a training method for LLMs that focuses on preserving the spectral properties of weight matrices to enhance training stability and generalization. The key idea is to reparameterize a weight matrix $\mathbf{W}$ as three matrices, $\mathbf{RWP}$, where $\mathbf{W}$ is a fixed matrix and $\mathbf{R}$ and $\mathbf{P}$ are learnable orthogonal matrices. This reparameterization ensures that the singular values of $\mathbf{W}$ are preserved throughout training. To improve the efficiency, the authors introduce two techniques: structured matrix sparsity (Stochastic Primitive Optimization) and efficient orthogonalization (Cayley-Neumann Parametrization). The experimental results show that the proposed method outperforms baseline methods such as AdamW, LoRA, and GaLore. Furthermore, the authors demonstrate that the proposed method is also effective when applied to fine-tuning.

**Questions:**

1. I would like to confirm that the parameters updated by the optimizer are $\mathbf{Q}$ in Eq. (6), not $\mathbf{G}$ itself.
2. Could you provide the training time of the proposed method?
3. Line 417: $R$ and $I$ should be in bold, as they represent matrices.

**Ethical Concerns:**

["NO or VERY MINOR ethics concerns only"]

**Final Justification:**

My concerns have been addressed. I will maintain my positive score.

**Limitations:**

yes

**Quality:**

3

**Strengths And Weaknesses:**

Strengths:
1. The proposed method preserves the spectral properties by the reparameterization and controls the spectral properties via matrix initialization. The authors investigate four types of initialization in their ablation studies.
2. The authors introduce a parameter-efficient sparse orthogonal matrix.
3. The authors present an efficient approximation method for orthogonalization, reporting a 1.5x speed-up (Appendix E).
4. The experiments are well-designed, and the ablation studies provide valuable insights.
5. The paper is clearly written.

Weaknesses:
1. The pretraining experiments report only perplexity, without any downstream task evaluation. Since pretrained models are typically used with fine-tuning or zero-/few-shot prompting, it might be beneficial to evaluate the pretrained models on downstream tasks as well.
2. The BOFT paper [49] presents a parameter-efficient sparse orthogonal matrix using butterfly factorization. Could this butterfly method be applied to POET in place of SPO? If so, it could serve as a reasonable baseline to assess the effectiveness of SPO.

---

> ### Author Rebuttal · Authors · 2025-07-31
>
> We sincerely thank **Reviewer jtaX** for the recognition of our contribution and the constructive comments on our work. We take every comment seriously and hope our response can address the reviewer’s concerns. If there are any remaining questions, we are happy to address them. We summarize all the concerns into the following questions:
>
> ---
>
> **Q1: The pretraining experiments report only perplexity, without any downstream task evaluation. Since pretrained models are typically used with fine-tuning or zero-/few-shot prompting, it might be beneficial to evaluate the pretrained models on downstream tasks as well.**
>
> A1: Great suggestion! To address the reviewer’s concerns, we have conducted the requested experiments to further demonstrate the effectiveness of POET-trained models.
>
> For comparison, we used the checkpoints of the 1.3B-AdamW baseline and our 1.3B-POET (FS, b=1/2) models from Table 2 of the main paper and evaluated them on the standard GLUE benchmark [1], which includes a diverse set of NLP tasks such as sentence classification, sentiment analysis, and question answering. For each task, we fine-tuned the models for 10 epochs using a consistent learning rate of 2e−5. The evaluation covers several finetuning strategies: full finetuning (FT) with AdamW, Low-Rank Adaptation (LoRA), and Orthogonal Finetuning (OFT), as well as finetuning with POET.
>
> Performance was measured using accuracy for MNLI, MRPC, QNLI, QQP, RTE, and SST-2; the Matthews Correlation Coefficient for CoLA; and the Pearson Correlation Coefficient for STS-B. The results, summarized in Table A (in the Rebuttal), show that the POET-pretrained model consistently outperforms the AdamW-pretrained baseline across all tasks and finetuning methods. Moreover, we find that POET-pretrained models, if also finetuned by POET, achieved the best overall downstream performance. These results further validate the effectiveness of our proposed method, and we will include these results, along with the table, in the revised version of our paper.
>
> **Table A. Downstream performance comparison on the GLUE benchmark between POET and AdamW-pretrained models.**
>
> | Metric | FT |  | LoRA|  |OFT |  | POET|  |
> |--|--|--|--|--|--|--|--|--|
> | Model  | 1.3B-AdamW | 1.3B-POET  | 1.3B-AdamW | 1.3B-POET  | 1.3B-AdamW | 1.3B-POET  | 1.3B-AdamW | 1.3B-POET  |
> | COLA   | 0.361      | **0.523**  | 0.423      | **0.460**  | 0.388      | **0.437**  | 0.435      | **0.505**  |
> | MNLI   | 0.658      | **0.818**  | 0.772      | **0.822**  | 0.774      | **0.812**  | 0.804      | **0.821**  |
> | MRPC   | 0.696      | **0.824**  | 0.689      | **0.760**  | 0.689      | **0.740**  | 0.806      | **0.826**  |
> | QNLI   | 0.818      | **0.885**  | 0.843      | **0.870**  | 0.842      | **0.855**  | 0.856      | **0.892**  |
> | QQP    | 0.829      | **0.902**  | 0.865      | **0.880**  | 0.867      | **0.877**  | 0.889      | **0.902**  |
> | RTE    | 0.534      | **0.661**  | 0.534      | **0.567**  | 0.531      | **0.538**  | 0.653      | **0.682**  |
> | SST2   | 0.914      | **0.920**  | 0.908      | **0.927**  | 0.915      | **0.924**  | 0.904      | **0.931**  |
> | STSB   | **0.880**  | 0.873      | 0.818      | **0.865**  | 0.762      | **0.791**  | 0.878      | **0.887**  |
>
> [1] Wang, Alex, et al. "GLUE: A multi-task benchmark and analysis platform for natural language understanding." arXiv preprint arXiv:1804.07461 (2018).
>
> ---
>
> **Q2: The BOFT paper [49] presents a parameter-efficient sparse orthogonal matrix using butterfly factorization. Could this butterfly method be applied to POET in place of SPO? If so, it could serve as a reasonable baseline to assess the effectiveness of SPO.**
>
> A2: Great suggestion! The butterfly factorization used in BOFT is a parameter-efficient orthogonal parameterization, and we agree with the reviewer that it can be a good way to further improve POET’s parameter-efficiency. We identified two primary ways to incorporate this factorization into POET training. The first is to directly replace our block-sparse SPO with a fixed butterfly factorization. While this approach offers better parameter efficiency, it involves additional matrix-matrix multiplications (butterfly factorization requires $\log(n)$ butterfly factors). This additional matrix-matrix multiplication, while being acceptable for finetuning (due to much shorter training runs), is computationally prohibitive for much longer pre-training schedules. Given this constraint, we could not feasibly train a model with this variant in the rebuttal period.
>
> Therefore, we implemented and tested the second variant. This approach adapts our "merge-then-reinitialize" step by replacing the random block permutation in POET-BS with a permutation ordered according to the butterfly factorization pattern. For a direct comparison, we trained two 60M parameter models on 5e10 tokens: one with our standard POET-BS and one with POET using this butterfly permutation. It effectively unrolls the butterfly factorization to different iterations of training. We ensured a fair comparison by using an identical parameter budget and the same set of hyperparameters for both methods.
>
> The results in Table B (in the Rebuttal) below indicate that standard POET-BS training achieves better performance. We hypothesize that this advantage stems from the increased stochasticity in standard POET-BS, which seems particularly beneficial during the exploratory phase of pretraining. Moreover, unrolling the butterfly factorization across different training iterations can be suboptimal, as the butterfly factors are not learned simultaneously. Each factor is updated in separate steps, potentially reducing overall expressiveness. Finally, while the BOFT framework excels in parameter efficiency, strong regularization, and its constrained orthogonal matrix structure, making it highly effective for finetuning tasks. In contrast, pretraining tasks, as explored in our paper, demand greater flexibility.
>
> We will include this new baseline comparison and the related discussion in the revised paper.
>
> **Table B. The validation perplexity comparison between POET and POET with the butterfly parameterization from the BOFT paper.**
>
> || POET-BS| POET-BS (Butterfly)|
> |--|--|--|
> | BS=256   | **25.75** | 26.73          |
> | BS=128   | **27.47** | 29.16          |
> | BS=64     | **31.82** | 32.93          |
>
> ---
>
> **Q3: I would like to confirm that the parameters updated by the optimizer are Q
>  in Eq. (6), not G itself.**
>
> A3: Yes, the reviewer’s understanding is correct. The parameters updated by the optimizers are the Q in Equation 6.
>
> ---
>
> **Q4: Could you provide the training time of the proposed method?**
>
> A4: Great suggestion! We are happy to perform the required training time comparison to address the reviewer’s concerns.
>
> To provide a fair and comprehensive comparison of training efficiency between POET and AdamW, we evaluate the training efficiency using two distinct metrics, following the experimental setup in Figure 9 of our paper. First, we report both the absolute clock time per iteration to measure computational overhead. Second, we compare the total time each method requires to reach an equivalent target validation perplexity, thereby assessing overall convergence speed. We note that, while POET has a higher cost per iteration due to the orthogonalization step, its overall time to a target performance may in fact be superior to standard AdamW training. By presenting both metrics, we offer a more complete and transparent assessment of our method's practical efficiency.
>
> Following the experimental setup in Figure 9 of the main paper, we report two metrics: iteration-wise training time (where higher iters/s indicates less time per iteration, so larger values are better) and total training time (where smaller values are better) required to reach approximately the same validation perplexity of 25.2 when training a Llama‑60M model. The results are summarized in the following tables:
>
> **Table C. Iteration-wise training time of POET.**
>
> |  | AdamW   | POET (FS, b=½) | POET (FS, b=1/4) |
> |--|--|--|-----|
> | Iteration-wise Training time   | 3.97 iters/s    | 3.27 iters/s     | 3.35 iters/s     |
>
> **Table D. Total training until reaching the same validation perplexity of 25.2.**
>
> |   | AdamW       | POET (FS, b=½) |
> |----|---|--|
> | Training time to same perplexity  | 50.4 h   | 23.8 h   |
>
> Indeed POET’s per-iteration training time is currently slower than AdamW’s. This overhead stems from several operations, including non-trivial matrix orthogonalization and sparse matrix multiplications, which involve advanced indexing and matrix permutations. While we have taken steps to reduce this overhead, such as implementing low-level optimizations with custom CUDA kernels and introducing the efficient Cayley–Neumann parameterization. But we do want to recognize that there remains room for further improvements in clock-time efficiency.
>
> We would also like to take this opportunity to clarify the scope of our work. Our primary goal is to demonstrate that the proposed principled energy/spectrum-preserving training paradigm can produce models that are consistently better than standard AdamW training up to a meaningful model scale (1.3B in the main paper, 3B in the rebuttal, see Table A in the response to Reviewer UGEa). At this stage, our focus has not been on heavily optimizing POET for large-scale, multi-node training, partly due to the resource constraints of our academic setting.
> However, we are confident that POET’s training efficiency can be significantly improved with more dedicated optimization to the low-level computations. We believe that improving POET’s efficiency and scalability represents a promising direction for future work.
>
> ---
>
> **Q5: Line 417: R and I should be in bold, as they represent matrices.**
>
> A5: Thanks for the remark! We have updated it in the revised paper.

---

> > ### Comment · Reviewer_jtaX · 2025-08-04
> >
> > Thank you very much for your time and effort in preparing the rebuttal. My concerns have been addressed. I will maintain my positive score.

---

> > > ### Author Response · Authors · 2025-08-04
> > > **Thank you!**
> > >
> > > Dear Reviewer,
> > >
> > > Thank you for your response and the valuable feedback provided in your review. We are pleased to have addressed your concerns and appreciate your constructive comments, which have helped improve our work.
> > >
> > > Best regards,

---

### Official Review · Reviewer_UGEa · 2025-07-04

**Clarity:** 3
**Significance:** 3
**Originality:** 3
**Rating:** 5
**Confidence:** 4

**Summary:**

The paper presents POET, a novel reparameterized training algorithm for LLMs that utilizes orthogonal equivalence transformation. By reparameterizing each neuron with two learnable orthogonal matrices and a fixed random weight matrix, POET preserves the spectral properties of weight matrices, stabilizing the optimization of the objective function and enhancing generalization. The authors also develop stochastic principal submatrix optimization and the Cayley-Neumann parameterization to improve POET's scalability. Experiments demonstrate POET's effectiveness across multiple model scales, including LLaMA models up to 1.3B parameters and BART models fine-tuned for summary generation.

**Questions:**

1. What are the bottlenecks in applying the current method to larger models such as those with 7B or 70B parameters?
2. What is the impact of the Cayley-Neumann iteration on training time and efficiency?
3. Does the Cayley-Neumann parameterization method encounter numerical stability issues when handling large-scale orthogonal matrices?

**Ethical Concerns:**

["NO or VERY MINOR ethics concerns only"]

**Final Justification:**

The authors addressed my concerns and questions and I still kept my previous scores.

**Limitations:**

Yes

**Quality:**

3

**Strengths And Weaknesses:**

Strengths:
The paper is technically sound with a solid theoretical foundation. The claims regarding POET's preservation of spectral properties are well-supported by experiments across multiple model scales and architectures. The authors demonstrate careful and honest evaluation of their work's strengths and limitations. The paper is clearly written and well-organized, offering a good understanding of the proposed method. It provides sufficient detail for expert readers to reproduce the results. The results are impactful for the LLM training community. POET addresses the challenge of training large models more stably and generalizably, making it likely for others to use or build on these ideas. The work provides new insights by leveraging orthogonal equivalence transformation for LLM training. It introduces a novel combination of existing techniques with clear articulation of the reasoning behind this combination.

Weaknesses:
The experiments only validate POET on models up to 1.3B parameters, leaving its effectiveness on significantly larger models unverified. The computational complexity analysis is limited, making it difficult to assess resource consumption in large-scale training scenarios. Some sections could benefit from additional clarity, particularly in explaining the Cayley-Neumann parameterization and its potential numerical stability issues. The significance is somewhat limited by the lack of validation on larger models. Addressing this would further enhance the paper's impact.

---

> ### Author Rebuttal · Authors · 2025-07-31
>
> We sincerely thank **Reviewer UGEa** for the recognition of our contribution and the constructive comments on our work. We take every comment seriously and hope our response can address the reviewer’s concerns. If there are any remaining questions, we are happy to address them. We summarize all the concerns into the following questions:
>
> ---
>
> **Q1: What are the bottlenecks in applying the current method to larger models such as those with 7B or 70B parameters? The experiments only validate POET on models up to 1.3B parameters, leaving its effectiveness on significantly larger models unverified. The computational complexity analysis is limited, making it difficult to assess resource consumption in large-scale training scenarios. Some sections could benefit from additional clarity, particularly in explaining the Cayley-Neumann parameterization and its potential numerical stability issues. The significance is somewhat limited by the lack of validation on larger models. Addressing this would further enhance the paper's impact.**
>
> A1: We appreciate the reviewer’s helpful remark and agree that additional scaling experiments would further strengthen the impact of our work. To address the reviewer’s concerns, we provide an experiment to demonstrate POET's scalability to 3B models (due to the short rebuttal period, we have to use less training tokens than our main paper). The experiment details are given later.
>
> First, we want to clarify that there are no fundamental limitations to applying POET to larger models. Because POET’s parameter/memory complexity depends on a hyperparameter $b$, which can usually be set to a value much smaller than $m,n$, we can control the computational budget required by POET. In our experiments, we have tested $b=1/8,1/4,1/2m$ for $R$ and $b=1/8,1/4,1/2n$ for $P$. However, POET with smaller memory budget (i.e., smaller b) will require longer iterations to reach the same performance, but it provides a useful way to trade-off performance and compute budget. In practice, we find that POET demonstrates better parameter-efficiency than standard AdamW training.
>
> While there are no fundamental limitations for scaling POET, training bigger models requires (1) additional engineering efforts; and (2) more GPU compute budget.
>
> Regarding (1), pretraining models beyond 7B parameters would necessitate optimizing the implementation for multi-node setups (e.g., inter-node communication overhead). However, because POET shares conceptual similarities with adapter-based finetuning methods, i.e., adding adapter weights to the base layers, a design known for its scalability. We anticipate straightforward scaling to models of 70B parameters and beyond given enough compute resources. Our current experiments support POET’s smooth scalability up to the 1.3B model. The experiments on 1.3B models reuse the same hyperparameters from that on much smaller models, as we do not have the resources to perform hyperparameter search for the larger  models. See the main results in Table 2 of the main paper.
>
> Regarding (2), our experiments are unfortunately constrained by the GPU budget of an academic lab, making it infeasible for us to scale POET to substantially larger models within the scope of this work.
>
> Although we do not have the necessary resources for the scaling experiment to 7B models, we address the reviewer’s concerns by scaling POET to the 3B model during this rebuttal phase. We find that training POET on 3B models still shows significant and consistent performance gains compared to standard training with AdamW.
>
> This experiment was designed to provide a direct comparison focused purely on scalability; therefore, we did not perform any hyperparameter tuning. Instead, we reused the exact hyperparameters from the 1.3B models reported in Table 2 of the main paper. We trained a 3B-parameter model on a single 8×H100 node with 5 billion training tokens using AdamW and POET (FS, b=1/2).
>
> We emphasize that, due to the time constraints of the rebuttal phase, we could not train the 3B model on the same number of tokens as in the main paper. Instead, we used only 1/10 of the training tokens used for the 1.3B model, which explains why the performance is slightly lower than that of the 1.3B model. The results, summarized in the following table, report the final validation perplexity for both methods. We observe that POET maintains a similar performance advantage over AdamW at the 3B scale, consistent with the findings shown in Table 2 from the main paper for smaller models. This confirms that POET’s benefits are not limited to the 1.3B scale but extend robustly to larger models.
>
> **Table A. Validation perplexity of training a Llama 3B model on 5 billion tokens.**
>
> |                                  | AdamW | POET(FS,b=1/2) |
> |----------------------------|-------------|-----------------------|
> | validation perplexity | 19.61      | 16.90                 |
>
> We hope these results address the reviewer’s concerns and will include them in the revised paper. We fully agree that demonstrating POET’s scalability to even larger models would further strengthen the impact of our work, and pursuing this will be a major direction for future research.
>
> ---
>
> **Q2: What is the impact of the Cayley-Neumann iteration on training time and efficiency?.**
>
> A2: Thank you for raising this question. We assume the question is about the number of Neumann series terms, we have included an ablation study in the paper (Table 5) to evaluate how varying the number of Neumann terms affects performance, using POET-FS with b=1/2 to train LLaMA-130M. We report the final validation perplexity and additionally the training time in the table below.
>
> **Table B: Ablation study: validation perplexity and training time with different numbers of Neumann series.**
>
> | Scheme   | Perplexity   | Time       |
> |----------|--------------|------------|
> |    k=1   | Not converged| 3.51 iters/s  |
> |    k=2   | 22.56        | 3.41 iters/s  |
> |    k=3   | 21.54        | 3.32 iters/s  |
> |    k=4   | 20.22        | 3.27 iters/s  |
> |    k=5   | 20.19        | 3.22 iters/s  |
>
> ---
>
> **Q3: Does the Cayley-Neumann parameterization method encounter numerical stability issues when handling large-scale orthogonal matrices?**
>
> A3: Thank you for raising this question. We understand the reviewer’s valid concern that performing orthogonalization on large matrices might encounter numerical stability issues. However, we did not encounter any numerical stability issues when dealing with large orthogonal matrices.
>
> Improving numerical stability is in fact one of the motivations for us to propose the Cayley-Neumann parameterization. To this end, SPO decomposes the training of large orthogonal matrices to many much smaller ones. In the Cayley-Neumann parameterization, Neumann series approximation gets rid of the matrix inverse, further enhancing its numerical stability.
>
> Moreover, we refer the reviewer to Table 2 of the main paper, where we compare the validation perplexity between POET and AdamW. For readability, the relevant results are summarized in Table C below. We find that POET with block-stochastic SPO demonstrates competitive performance against AdamW across different model scales. The key advantage of this approach is that the block size for orthogonalization is fixed (e.g., 256×256 in our settings) and does not grow with the layer dimension. Larger models can be simply trained with more of these small, fixed-size orthogonal matrices. Therefore, numerical stability is not a problem here. Specifically for the Neumann series, as long as we set the number of terms $k\geq 3$, we never encounter any numerical stability issues in practice.
>
>
> **Table C. Validation perplexity comparison of POET-BS and AdamW.**
>
> | Model (# tokens)       | 60M (30B)               | 130M (40B)              | 350M (40B)              | 1.3B (50B)             |
> |-----------------------|-------------------------|-------------------------|-------------------------|------------------------|
> | AdamW                 | 26.68 (25.30M)          | 20.82 (84.93M)          | 16.78 (302.38M)         | 14.73 (1.21B)          |
> | POET(BS,b=256) | **25.29** (9.66M)       | **19.88** (22.33M)      | **16.27** (60.32M)          | **14.56** (118.26M)        |

---

> ### Comment · Area_Chair_22zL · 2025-08-08
> **Please review authors' response**
>
> Dear Reviewer UGEa,
>
> According to the conference policy, the reviewer must be involved in the author-reviewer discussion.
>
> Could you please review the author's response and check whether they have addressed your concerns?
>
> Thank you.
>
> AC

---

### Official Review · Reviewer_sLfu · 2025-07-08

**Clarity:** 2
**Significance:** 3
**Originality:** 3
**Rating:** 4
**Confidence:** 2

**Summary:**

This paper proposes to train deep learning models with a novel method POET based on orthogonal reparameterization. Specifically, it periodically reinitialize the weight matrix $W$ to $RWP$, where $R$ and $P$ are orthogonal matrices. To further improve efficiency, this paper considers two distinct strategies—Full stochastic SPO and block-stochastic SPO—to break down the full orthogonal matrix into smaller pieces, and make them share parameters to further improve parameter efficiency. To improve numarical stability, the authors use a truncated Neumann series to approximante the matrix inverse operation inherited in the Cayley-Neumann parameterization for constructing orthogonal matrices. By periodically merging $R$ and $P$ into $W$ and reinitializing $W$ to $RWP$, the operator norm can be effectively controlled to guarantee such approximations. Besides, the paper has conducted experimental studies on pre-training LLaMA models, exhibiting advantages on both efficiency and training performance.

**Questions:**

What is the major conclusion drawn from the empirical observations of the three learning stages? How can it help with developing the POET method, or justifying the use of it?

**Ethical Concerns:**

["NO or VERY MINOR ethics concerns only"]

**Final Justification:**

I have no further concerns or questions, and I will maintain my score as is.

**Limitations:**

Yes.

**Quality:**

3

**Strengths And Weaknesses:**

Strengths:

i) The technical details, particularly parameterization details and training procedures, are novel and interesting.

ii) The potential of this method is promising. With fewer trainable parameters, the method can achieve comparable performance and superior training efficiency.

Weaknesses:

i) The trade-off between training efficiency and training performance is not super clear. Although POET are more parameter-efficient, most training curves in the paper have shown that Adam is still much faster (2~3 times, roughly) than POET in the initial training stages. Consequently, it is unclear if we should use Adam to train in the early stages for faster convergence.

ii) Technical details missing. Maybe I missed it but it remains unclear to me whether the optimizer needs restart whenever the weights are merged and reinitialized.

iii) The experimental setup is insufficient. First, the models used are not bigger than 1.3B. It is recommended to include 7B results at least for fine-tuning tests. Second, given the motivation to enhance generalization ability by controlling the spectral norm, I expect to see more results illustrating the generalization ability of POET-trained models, which typically requires accuracy scores on downstream tasks such as GLUE.

---

> ### Author Rebuttal · Authors · 2025-07-31
>
> We sincerely thank **Reviewer sLfu** for the recognition of our contribution and the constructive comments on our work. We take every comment seriously and hope our response can address the reviewer’s concerns. If there are any remaining questions, we are happy to address them. We summarize all the concerns into the following questions:
>
> ---
>
> **Q1: The trade-off between training efficiency and training performance is not super clear. Although POET are more parameter-efficient, most training curves in the paper have shown that Adam is still much faster (2~3 times, roughly) than POET in the initial training stages. Consequently, it is unclear if we should use Adam to train in the early stages for faster convergence.**
>
> A1: Great question! The reviewer’s main concern is that, based on the perplexity curves, AdamW appears to converge faster initially, raising doubts about whether POET can sustain its strong performance with fewer training steps.
>
> We note that this convergence pattern reflects differences in training dynamics, particularly how each method responds to the learning rate scheduler, rather than demonstrating superior efficiency for AdamW. Specifically, POET exhibits distinct training phases, as illustrated in Figure 2 from the main paper. As long as the same learning rate scheduler is applied to both AdamW and POET, POET can still outperform AdamW even with significantly fewer training steps.
>
> To fully address the reviewer’s concerns, we conducted an ablation study on the Llama-60M model, reducing the training to 1/6 and 1/3 of the total steps reported in Table  A (in the Rebuttal), while keeping all other hyperparameters unchanged.
>
> The results show that even with significantly fewer training steps, POET consistently achieves a lower final perplexity than AdamW. This demonstrates that POET’s effectiveness is not limited to long training runs and that AdamW does not converge faster than POET, even under shorter training schedules. That said, we indeed find that the performance gap between POET and AdamW tends to widen with longer training steps, which aligns with the findings reported in our main paper.
>
> **Table A. Results of much fewer training steps for the 60M model.**
>
> | Training Steps | AdamW | POET (FS, b=½) |
> | :------------- | :---: | :--------------: |
> | 50,000         | 28.72 |      **28.68** |
> | 100,000        | 27.38 |      **27.03** |
> | 300,000 (Paper)| 26.68 |      **25.37** |
>
> Additionally, we would like to share some insights on combining AdamW and POET training. We refer the reviewer to Table B (in the Rebuttal), which presents results on mixing strategies, such as pretraining with AdamW and then continuing (finetuning) with POET on downstream tasks, or vice versa. Our observations show no fundamental limitations in mixing AdamW and POET: a model pretrained with POET can be further trained with AdamW, and vice versa, while still achieving strong downstream performance. However, we find that using POET throughout the entire training always yields better performance. More importantly, using this mixed training strategy will destroy many nice properties such as weight spectrum and energy preservation. This may also be the reason that the mixed training strategy performs worse than using POET alone.
>
>
>
> ---
>
> **Q2: Technical details missing. Maybe I missed it but it remains unclear to me whether the optimizer needs restart whenever the weights are merged and reinitialized.**
>
> A2: Thanks for the question. Yes, after performing the merge-then-reinitialize trick, we need to reset the optimizer states since the parameters being updated have changed. In developing this method, we also experimented with merge-then-reinitialize without resetting the optimizer states, but consistently observed worse performance.
>
> We also note that this type of momentum reset has been used in prior methods [1] and does not harm final performance. We will add further explanation to the merge-then-reinitialize trick section in the revised paper to clarify this point.
>
> [1] Huang, et al. "SPAM: Spike-aware adam with momentum reset for stable LLM training." ICLR 2025.
>
> ---
>
> **Q3: The experimental setup is insufficient. First, the models used are not bigger than 1.3B. It is recommended to include 7B results at least for fine-tuning tests. Second, given the motivation to enhance generalization ability by controlling the spectral norm, I expect to see more results illustrating the generalization ability of POET-trained models, which typically requires accuracy scores on downstream tasks such as GLUE..**
>
> A3: Great suggestion! We agree with the reviewer that whether a base model is good or not depends on its downstream performance. To address the reviewer’s concerns, we have conducted all the required experiments to further demonstrate the effectiveness of our proposed method.
>
> **Performance on downstream tasks**: We start by showing the results on the requested downstream task GLUE [2].
>
> For comparison, we took the checkpoints of the 1.3B-AdamW baseline and our 1.3B-POET (FS, b=1/2) from Table 2 of the main paper and evaluated them on the standard GLUE benchmark [1]. For each task, we finetuned the models for 10 epochs using a consistent learning rate of 2e−5. The evaluation covers several finetuning strategies: full finetuning (FT) with AdamW, Low-Rank Adaptation (LoRA), and Orthogonal Finetuning (OFT), as well as finetuning with POET.
>
> Performance was measured using accuracy for MNLI, MRPC, QNLI, QQP, RTE, and SST-2; the Matthews Correlation Coefficient for CoLA; and the Pearson Correlation Coefficient for STS-B. The results, summarized in the Table B (in the Rebuttal) below, show that the POET-pretrained model consistently outperforms the AdamW-pretrained baseline across all tasks and finetuning methods. This analysis further validates the effectiveness of POET, and we will include all these results in the revised paper.
>
> **Table B: Downstream performance comparison on the GLUE benchmark between POET- and AdamW-pretrained models.**
>
> | Metric | FT         |         | LoRA       |       | OFT        |        | POET       |       |
> |--------|------------|------------|------------|------------|------------|------------|------------|------------|
> | Model  | 1.3B-AdamW | 1.3B-POET  | 1.3B-AdamW | 1.3B-POET  | 1.3B-AdamW | 1.3B-POET  | 1.3B-AdamW | 1.3B-POET  |
> | COLA   | 0.361      | **0.523**  | 0.423      | **0.460**  | 0.388      | **0.437**  | 0.435      | **0.505**  |
> | MNLI   | 0.658      | **0.818**  | 0.772      | **0.822**  | 0.774      | **0.812**  | 0.804      | **0.821**  |
> | MRPC   | 0.696      | **0.824**  | 0.689      | **0.760**  | 0.689      | **0.740**  | 0.806      | **0.826**  |
> | QNLI   | 0.818      | **0.885**  | 0.843      | **0.870**  | 0.842      | **0.855**  | 0.856      | **0.892**  |
> | QQP    | 0.829      | **0.902**  | 0.865      | **0.880**  | 0.867      | **0.877**  | 0.889      | **0.902**  |
> | RTE    | 0.534      | **0.661**  | 0.534      | **0.567**  | 0.531      | **0.538**  | 0.653      | **0.682**  |
> | SST2   | 0.914      | **0.920**  | 0.908      | **0.927**  | 0.915      | **0.924**  | 0.904      | **0.931**  |
> | STSB   | **0.880**  | 0.873      | 0.818      | **0.865**  | 0.762      | **0.791**  | 0.878      | **0.887**  |
>
> **Performance on a larger model**: We now present results on pretraining with POET and AdamW at a larger model scale. Since this research is conducted in an academic setting, our computing resources are limited. As a result, the largest model we could feasibly pretrain for a controlled comparison within the short rebuttal period was a 3B-parameter model.
>
> To this end, we trained a 3B-parameter model on a single 8×H100 node using 5 billion tokens with both AdamW and POET (FS, b=1/2). We emphasize that, due to the time constraints of the rebuttal phase, we were unable to train the 3B model on the same number of tokens as in the main paper. Instead, we used only 1/10 of the training tokens used for the 1.3B model, which explains why its performance is slightly lower than that of the 1.3B model. It is important to note that, to ensure a direct comparison focused purely on scalability, we did not perform any hyperparameter tuning. We simply reused the exact hyperparameters applied to the 1.3B models reported in Table 2 of the main paper.
>
> The results, summarized in Table C (in the Rebuttal), report the final validation perplexity for both methods. We observe that POET maintains a similar performance advantage over AdamW at the 3B scale, consistent with the results shown in Table 2 of the main paper for smaller models. This demonstrates that POET’s advantages are not limited to the 1.3B scale but instead exhibit robust scalability to larger models.
>
> **Table C. Validation perplexity of training a Llama 3B model on 5 billion tokens.**
>
> |                                  | AdamW | POET(FS,b=1/2) |
> |----------------------------|-------------|-----------------------|
> | validation perplexity | 19.61      | 16.90                 |
>
>
> [2] Wang, et al. "GLUE: A multi-task benchmark and analysis platform for natural language understanding." arXiv preprint arXiv:1804.07461.

---

> > ### Comment · Reviewer_sLfu · 2025-08-01
> >
> > Thanks for the rebuttal. The points make sense to me and I have no further questions.

---

> > > ### Author Response · Authors · 2025-08-01
> > > **Thank you!**
> > >
> > > Dear Reviewer,
> > >
> > > Thank you for your response and the valuable feedback provided in your review. We are pleased to have addressed your concerns and appreciate your constructive comments, which have helped improve our work.
> > >
> > > Best regards,

---

### Official Review · Reviewer_hgyC · 2025-07-22

**Clarity:** 3
**Significance:** 3
**Originality:** 2
**Rating:** 4
**Confidence:** 2

**Summary:**

This paper is  concerned with stable and efficient training of LLMs by means of imposing orthogonality on  weights matrices. The orthogonal weight training for DNNs is already used in the literature to  improve generalization. The  two initilization schemes for orthogonal weight training is novel. Numerical results indicate that, when AdamW is trained with nearly three times more tokenswithout repeating any and using a well-tuned learning rate—the improvement is significant and not just due to more training steps. Additionally, results show that POET's performance scales with parameter budget: larger models consistently perform better, which has key implications for scaling laws.

**Questions:**

The connection of  spectral norm of weights  to generalization bound that you used in (7) is for MLP, is the connection to attention-based  architecture  studied in the literature or a direct result of  (7), please  clarify.

**Ethical Concerns:**

["NO or VERY MINOR ethics concerns only"]

**Final Justification:**

The authors  provided a detail rebuttal and clarified that some of the missing proofs are already in he literature, which I hope they will  be  properly cited and  explained in the paper.I am satisfied with the replies and explains and will keep my score.

**Limitations:**

The proofs in the supplementary meterial  only cover gaussian  initialization and SPO and Cayley-Neumann parametrization  are not proved to be affecting the generalization,

**Paper Formatting Concerns:**

The main theorems  in the supplementary material should be move to the main body of the paper for better  understanding.

**Quality:**

3

**Strengths And Weaknesses:**

The Streght of  this paper is that,  extensive  numerical simulations are provided  to demonstrate the  effectiveness of orthogonal training in LLMs. Weaknesses are:
- The inductive bias of POET mentioned on page  7 is not clearly explained. As a reader I can not determine what is the inductive bias the POET has. This paragraph needs ellaboration or mathematical proof.
- No mathematical proof is  provided on  how the  SPO and CNP initialization affect the generalization bound. Moreover, the  main generalization bound used in Th.1 is for MLP and not an LLM.

---

> ### Author Rebuttal · Authors · 2025-07-31
>
> We sincerely thank **Reviewer hgyC** for the recognition of our contribution and the constructive comments on our work. We take every comment seriously and hope our response can address the reviewer’s concerns. If there are any remaining questions, we are happy to address them. We summarize all the concerns into the following questions:
>
> ---
>
> **Q1: The inductive bias of POET mentioned on page 7 is not clearly explained. As a reader I can not determine what is the inductive bias the POET has. This paragraph needs ellaboration or mathematical proof.**
>
> A1: Thanks for the question. To be clear, the inductive bias we discussed here refers to constraints on the parameterized function. The unique weight update mechanism of POET introduces a few interesting explicit and implicit inductive biases, as detailed below. To address the reviewer’s concerns, we will add more context to better explain the inductive bias discussion.
>
> For explicit inductive biases:
>
> **Spectrum-preserving weight matrices**: previous work [1] has shown that AdamW suffers from the lack of spectral diversity of the weight update, and gradient matrices, which have a more diverse spectrum, leads to favorable performance. We performed the same spectral analysis, by calculating the SVD entropy of the weight matrices during the training of a Llama 60M model and reporting the results in the following table. We show that POET, by guaranteeing to preserve the weight spectrum from the initialization, consistently maintains high spectral diversity throughout training, and therefore, POET can better explore diverse optimization directions.
>
>
> **Table A. Comparison of SVD entropy between a AdamW-pretrained model and a POET-pretrained model.**
>
> | Layer     | Initialization | AdamW  | POET   |
> |-----------|----------------|--------|--------|
> | Q proj    | 0.92           | 0.865  |**0.92**|
> | K proj    | 0.92           | 0.861  |**0.92**|
> | Up proj   | 0.97           | 0.946  |**0.971**|
> | Down proj | 0.97           | 0.861  |**0.972**|
> | Gate proj | 0.97           | 0.93  |**0.971**|
> | Out proj  | 0.92           | 0.897  |**0.92**|
>
> **(Probabilistic) energy-preserving weight matrices**: In addition, we compare the hyperspherical energy [2] of the model trained with AdamW and with POET. Hyperspherical energy, as introduced in our paper, is a quantity that characterizes the uniformity of neurons on the unit hypersphere.
>
> The results are summarized in the Table B (in the Rebuttal)  below. Compared to AdamW, POET provides a much stronger guarantee of minimizing hyperspherical energy throughout training. It is important to note that the absolute value of the energy does not directly characterize neuron uniformity; rather, it is the relative difference from the minimum energy that reflects uniformity. Moreover, the initialization energy is already very close to the minimum energy (owing to the properties of the zero-mean isotropic Gaussian initialization).
>
> **Table B. Comparison of hyperspherical energy between a AdamW-pretrained model and a POET-pretrained model.**
>
> |     | Initialization | AdamW  | POET   |
> |-----------|----------------|--------|--------|
> | Energy    | 39.616          | 39.644  |**39.615**|
>
> The energy difference between AdamW and initialization is about 0.03—roughly 30 times larger than the difference between POET and initialization. This gap is substantial and signals a significant deviation from uniformity, even though the raw values may appear small due to the high dimensionality. Notably, POET achieves an energy level almost identical to the initialization, demonstrating its energy-preserving (and energy-minimizing) inductive bias.
>
> For implicit inductive biases, the multiplicative weight update can be viewed as a long sequence of matrix factorization of the final weight matrix. Specifically, if rolled out across multiple iterations, the final trained weight matrix can be equivalently expressed by $R_n \cdots R_1 W_0 P_1 \cdots P_n$ where $R_i$ and $P_i$ are orthogonal matrices. It may have some implicit inductive biases, as suggested by [3,4,5]
>
> [1] Liu, et al. Muon is scalable for LLM training. arXiv preprint arXiv:2502.16982 (2025).
>
> [2] Liu, et al. Learning towards minimum hyperspherical energy. NeurIPS 2018
>
> [3] Gunasekar, et al. Implicit regularization in matrix factorization. NeurIPS 2017
>
> [4] Arora, et al. Implicit regularization in deep matrix factorization. NeurIPS 2019
>
> [5] Li, et al. Algorithmic Regularization in Over-parameterized Matrix Recovery and Neural Networks with Quadratic Activations. COLT 2018
>
> ---
>
> **Q2: No mathematical proof is provided on how the SPO and CNP initialization affect the generalization bound.**
>
> A2: Thanks for the question. Recall that we consider the initialization $W_{RP}=RW_0 P$ where $R$ and $P$ are initialized as identity matrices and $W_0$ follows a standard initialization scheme. Therefore, in terms of initialization, there is no difference between POET and standard training.
>
> In particular, the spectral norms of our methods at initialization are exactly the same as standard training, so our method enjoys the same generalization guarantees in Eq.(7) of the main paper.
> SPO then refers to our approximation that we do not optimize over $R$ directly but only over suitable sub-blocks. We note that SPO does not compromise the orthogonality. Therefore, even with SPO, the spectral norm of weight matrices will not change, validating the generalization bound.
>
> CNP is an approximation scheme that avoids expensive matrix inversion and approximates the orthogonal matrix $R$ by a power series. Since CNP does not result in perfectly orthogonal matrices, the singular values of $W_{RP}$ change slightly but this change can be bounded by controlling the remainder of the Neumann series and Figure 1(b) from the main paper shows that it is very small in practice (almost neglectable). Therefore the generalization bounds do not deteriorate during training.
>
> However, we note that the generalization bound serves as an intuitive understanding of why controlling spectral norm is beneficial to generalization. There are no theoretical guarantees that can give a rigorous and tight generalization bound for practical LLMs. Therefore, a more involved theoretical analysis is beyond the scope of this paper. We do agree with the reviewer that more theoretical understanding will be useful and should be an important future direction.
>
> ---
>
> **Q3: The connection of spectral norm of weights to generalization bound that you used in (7) is for MLP, is the connection to attention-based architecture studied in the literature or a direct result of (7), please clarify.**
>
> A3: Thanks for the question. This is a good point. At present, we cite well-known generalization bounds for MLPs to motivate the idea that controlling spectral norms generally supports generalization. To strengthen this argument, we will add references to additional works demonstrating similar results for transformer-based models. In fact, [1] and its extension [2] derive generalization bounds for transformer models that also involve the spectral norms of the weight matrices.
>
>
> [1] Edelman, et al.. Inductive Biases and Variable Creation in Self-Attention Mechanisms. ICML 2022
>
> [2] Trauger, et al. Sequence Length Independent Norm-Based Generalization Bounds for Transformers. AISTATS 2024
>
>
> ---
>
> **Q4: the proofs in the supplementary material only cover gaussian initialization and SPO and Cayley-Neumann parametrization are not proved to be affecting the generalization.**
>
> A4: Thanks for the question. The statements on the generalization bounds are based on the norms of the weight matrices.  SPO does not change the norm because it only restricts the optimization to a subset of all orthogonal matrices. The CNP approximation results in small changes in the spectrum which can be bounded by $O(|Q|^{k+1})$ where $|Q|$ remains small due to the merge and reinitialize trick. We will add more discussion on this to address the reviewer’s concerns.
>
> More importantly, we want to emphasize that our contribution does **NOT** lie in establishing a new generalization bound (nor did we make such a claim in the paper). Instead, we use existing generalization bounds as intuitive motivation and justification for why controlling spectral norms can be beneficial. Providing a rigorous theoretical generalization analysis for LLM training is beyond the scope of this work. Our contribution is the proposed reparameterization of weight matrices and the corresponding training algorithm for LLMs.

---

> > ### Comment · Reviewer_hgyC · 2025-08-08
> >
> > Thank you for the thorough repsonse, I will retain my positive evaluation.

---

### Comment · Area_Chair_22zL · 2025-08-04
**Reminder: Please Review Author Responses**

Dear Reviewers,

As the author-reviewer discussion deadline approaches, I would like to remind you to please carefully review the author responses. This rebuttal phase is a critical opportunity for authors to clarify misunderstandings, address concerns, and provide supplementary evidence for their work.

When evaluating the rebuttals, please consider the authors' arguments and assess whether your initial concerns have been satisfactorily addressed.

Your thorough engagement in this dialogue is vital for a fair and constructive review process and is essential for upholding the high standards of NeurIPS. Thank you for your dedication and expertise.

Best regards,
Area Chair

---

### Author Response · Authors · 2025-08-08
**Thank you!**

Dear Reviewers and Area Chair,

Thank you for your valuable time and insightful feedback on our submission. We appreciate the responsible and constructive nature of your reviews. We are pleased that we were able to address all the concerns raised. We will ensure that the required changes are incorporated into the revised version.

Best,
The authors

---

### Note · Authors · 2025-08-14

We would like to take this final opportunity to thank all the reviewers and the AC for their time and effort in evaluating our submission. In recognition of the reviewers’ efforts, we have conducted all the requested experiments and carefully addressed every concern raised. We deeply appreciate the reviewers’ active engagement during the rebuttal stage, and we are very glad that all reviewers are satisfied with our responses.

We want to also briefly highlight a few new results suggested by the reviewers:

- POET demonstrates significant performance gain under an increased model size (3B), indicating that its effectiveness remains consistent as models scale.

- POET-pretrained models not only perform better in terms of validation perplexity, but also achieve consistently better downstream performance under multiple finetuning schemes (LoRA, OFT, Full finetuning, POET finetuning).

- To reach the same target loss, POET requires substantially less training time than standard AdamW training, demonstrating its potential for efficient and scalable pretraining.

We will incorporate the reviewers’ constructive suggestions and include all additional experimental results from the rebuttal stage in the revised paper.

---

### Decision · Program_Chairs · 2025-09-17

**Decision:**

Accept (poster)

**Comment:**

This work introduces POET, a reparameterized training algorithm for large language models that optimizes neurons through orthogonal equivalence transformations. Each neuron is reparameterized with two learnable orthogonal matrices and a fixed random weight matrix, preserving the spectral properties of weight matrices to enable stable optimization and improved generalization. Efficient approximations further make the method scalable to large networks, and experiments validate its effectiveness for LLM training.

During the rebuttal, the authors provided sufficient evidence and clarification, demonstrating that POET scales effectively to larger models (3B), achieves lower validation perplexity, delivers superior downstream performance across multiple finetuning methods, and converges faster than AdamW. These results convincingly address the reviewers’ concerns.

**Recommendation**: Accept.